# Functional recruitment of dynamin requires multimeric interactions for efficient endocytosis

Morgane Rosendale [1,2,5,7], Thi Nhu Ngoc Van [1,2,6,7], Dolors Grillo-Bosch [1,2,7], Silvia Sposini [1,2], Léa Claverie[1,2], Isabel Gauthereau [1,2], Stéphane Claverol[3], Daniel Choquet [1,2,4], Matthieu Sainlos [1,2]* & David Perrais [1,2]*

During clathrin mediated endocytosis (CME), the concerted action of dynamin and its interacting partners drives membrane scission. Essential interactions occur between the proline/arginine-rich domain of dynamin (dynPRD) and the Src-homology domain 3 (SH3) of various proteins including amphiphysins. Here we show that multiple SH3 domains must bind simultaneously to dynPRD through three adjacent motifs for dynamin's efficient recruitment and function. First, we show that mutant dynamins modified in a single motif, including the central amphiphysin SH3 (amphSH3) binding motif, partially rescue CME in dynamin triple knock-out cells. However, mutating two motifs largely prevents that ability. Furthermore, we designed divalent dynPRD-derived peptides. These ligands bind multimers of amphSH3 with >100-fold higher affinity than monovalent ones in vitro. Accordingly, dialyzing living cells with these divalent peptides through a patch-clamp pipette blocks CME much more effectively than with monovalent ones. We conclude that dynamin drives vesicle scission via multivalent interactions in cells.

[1] University of Bordeaux, F-33000 Bordeaux, France. [2] CNRS, Interdisciplinary Institute for Neuroscience, UMR 5297, F-33000 Bordeaux, France. [3] Proteome Platform, Functional Genomic Center of Bordeaux, University of Bordeaux, Bordeaux, France. [4] Bordeaux Imaging Center, UMS 3420 CNRS, Université de Bordeaux, US 4 INSERM, F-33000 Bordeaux, France. [5] Present address: CNRS, Institut des Sciences Moléculaires, UMR 5255, 33405 Talence, France. [6] Present address: Sys2diag, Montpellier, France. [7] These authors contributed equally: Morgane Rosendale, Thi Nhu Ngoc Van, Dolors Grillo-Bosch. *email: matthieu.sainlos@u-bordeaux.fr; david.perrais@u-bordeaux.fr

Clathrin mediated endocytosis (CME) is fundamental to all eukaryotic cells. It is implicated in nutrient uptake, cell motility, cell signalling and neurotransmission[1,2]. The formation of clathrin coated vesicles (CCVs) from the plasma membrane involves more than 50 proteins, each recruited in specific steps. Multi-colour fluorescent live cell imaging of reference proteins, including clathrin, a constitutive cargo such as the transferrin receptor (TfR), as well as other associated proteins, has unveiled a stereotyped sequence of protein recruitments leading to CCV formation[3,4]. This choreography is orchestrated by a dense network of protein-protein and protein-lipid interactions, which, taken individually, are rather weak. This initially surprising property has been proposed to ensure the directionality and irreversibility of the process[5]. Therefore, understanding how this intricate network is organized and regulated in space and time is a major challenge of the field.

The large GTPase dynamin is essential for vesicle scission[6] and is primarily recruited at the last steps of vesicle formation[4,7,8]. Many aspects of the function of this indispensable protein have been deciphered[9] but how its recruitment is regulated in cells is still a matter of debate. In this regard, the C-terminal proline/arginine rich domain of dynamin (dynPRD) is critical. It is a major interaction hub for Src-homology domain 3 (SH3) domain-containing endocytic accessory proteins[10] that bears several class I (RxxPxxP) and class II (PxxPxR) SH3-binding motifs (where P is proline, R is positively charged arginine and x is any amino-acid)[11]. These motifs interact with various specificities to many SH3 domain containing proteins involved in CME such as intersectin1-2[12], endophilinA1-3[13], amphyphysin1-2[14], syndapin1-3[15], SNX9 and SNX18[16]. Among those, amphiphysins play a central role in regulating the recruitment of dynamin at CME sites[17–21]. The amphiphysin specific binding site on dynPRD, conserved in all three dynamin isoforms, consists of two overlapping class-II SH3-binding motifs resulting in its characteristic PxRPxR consensus sequence[14]. A so-called D15 peptide containing this particular motif has been extensively used to inhibit endocytosis in living neuronal cells[18,22–24]. However, its low affinity for the SH3 domain of amphiphysin (amphSH3), 90 μM[22], makes it a poor competitor of endogenous dynamin. This suggests that the specific binding of a single SH3 to a given motif on dynPRD cannot solely account for the functional interaction of dynamin with its associated proteins and that avidity effects may be at play in vivo.

The essential role of dynamin in CME, as well as the wealth of information about its recruitment to nascent endocytic vesicles and its interactions with SH3 domain containing proteins, makes the dynPRD-SH3 interaction network a very good candidate to test the hypothesis that weak individual interactions are reinforced by avidity effects in the scission machinery in vivo. To do so, we set up two complementary experimental approaches. First, we rescue CME in dynamin triple knock-out (TKO) cells[25] by re-expressing a number of dynamin mutants. We show that multiple neighbouring SH3 binding motifs in the dynPRD are necessary for CME rescue and for the timely recruitment of dynamin to nascent CCVs. Second, we develop divalent peptide-based amphSH3 ligands that are designed to mimic the native interactions and bind multimeric amphSH3 domains with high affinity. To test these peptides in living cells, we set up a protocol that combines the ppH assay, which detects the formation of endocytic vesicles with high temporal resolution[4,26], with patch-clamp mediated cell dialysis. We show that at a given concentration, these peptides block CME in living cells within minutes more efficiently than the commonly used monovalent D15 peptide. These results demonstrate the importance of multimeric interactions and reveal avidity effects in regulating the scission machinery.

## Results

**Multiple SH3 binding sites support dynamin function in cells**. We tested which feature of the PRD domain of dynamin is necessary for its optimal function in CME. We based our approach on rescuing CME in dynamin TKO mouse embryonic fibroblast (MEF) cells by re-expressing wild-type or mutant dynamin2 tagged with GFP (Fig. 1a). Indeed, unlike the neuronally enriched dynamins 1 and 3, the ubiquitously expressed dynamin 2 was reported to fully rescue CME in dynamin1,2 KO cells[27]. First, we confirmed the knock-out of endogenous dynamins and re-expression of dyn2-GFP to comparable levels by Western blot (Fig. 1b, Supplementary Fig. 1A). We also validated the ability of dyn2-GFP to rescue CME by monitoring the uptake of Alexa568 labelled transferrin (A568-Tfn). Wild-type dyn2-GFP fully rescued the uptake of A568-Tfn (103.7 ± 7.5% of control, $p > 0.99$), normally abolished in TKO cells (22.8 ± 1.6%, $p < 0.0001$, Fig. 1c, e). We additionally verified that dyn2-GFP expression levels did not influence A568-Tfn internalisation (Supplementary Fig. 1B) such that we could pool the data obtained with a given construct and use as a single dataset. Then, we tested dyn2-GFP mutated in the C-terminal portion of the PRD according to our hypothesis. We had noted that this domain contains three class II SH3 binding motifs. We labelled them A, B and C such that motif B is part of the known D15 peptide while motifs A and C flank motif B as illustrated in Fig. 1d. Of note, all dynamin mutants tested thereafter were expressed in TKO cells at similar levels (Supplementary Fig. 1C). As expected[6], neither re-expression of dyn2-GFP lacking the whole PRD domain (dyn2-GFP-ΔPRD) nor the D15-containing C-terminal domain (dyn2-GFP-ΔCter) could rescue endocytosis in those cells (Fig. 1e). But interestingly, the same phenotype was obtained with the much milder mutant dyn2-GFP-ABC$_{mut}$ in which only the 6 arginine residues present in motifs A, B and C were mutated to alanine (Fig. 1e). On the other hand, mutating motif B alone (dyn2-GFP-B$_{mut}$) resulted in a partial rescue of endocytosis (Fig. 1e). This suggests that the flanking binding motifs A and/or C play a role in the rescue process, potentially by stabilizing the PRD-SH3 interaction. To test this hypothesis further we mutated these motifs individually. Mutating flanking motif A alone (dyn2-GFP-A$_{mut}$) permitted a full rescue of CME. Mutating flanking motif C alone (dyn2-GFP-C$_{mut}$) allowed a partial rescue, to a similar extent as when only the core motif B was mutated. These results thus reveal that motif B is not the sole determinant of dynamin function in CME. Consistent with these results on Tfn uptake, mutant dyn2-GFP localisation in cells was very diverse, from punctate for the endocytically active mutants (dyn2-GFP and dyn2-GFP-A$_{mut}$) to homogenous for the inactive mutants (dyn2-GFP-ΔPRD/ΔCter/ABC$_{mut}$) and intermediate for the partial mutants (Fig. 1f). Hence our working model: binding of amphiphysin to the PRD requires the core B motif as well as another motif for stabilisation. Motif C is the prime target when available. However, when this site is not available, motif B appears to be stabilized enough to impart a partial rescue. The prime candidate for this partial stabilisation is the flanking motif A, which when mutated alone appeared to play no role. We confirmed this hypothesis by mutating motifs A and C simultaneously (dyn2-GFP-AC$_{mut}$). Under these conditions, the rescue of endocytosis was very poor (albeit significantly above the inactive mutants dyn2-GFP-ΔPRD, ΔCter and ABC$_{mut}$). Again, this confirms that the B (D15) motif alone is not sufficient to ensure the full functional recruitment of dynamin.

**Recruitment kinetics of dynamin PRD mutants**. To understand at which stage of CME dynamin function was affected by PRD mutations, we analysed their recruitment using TIRF imaging in

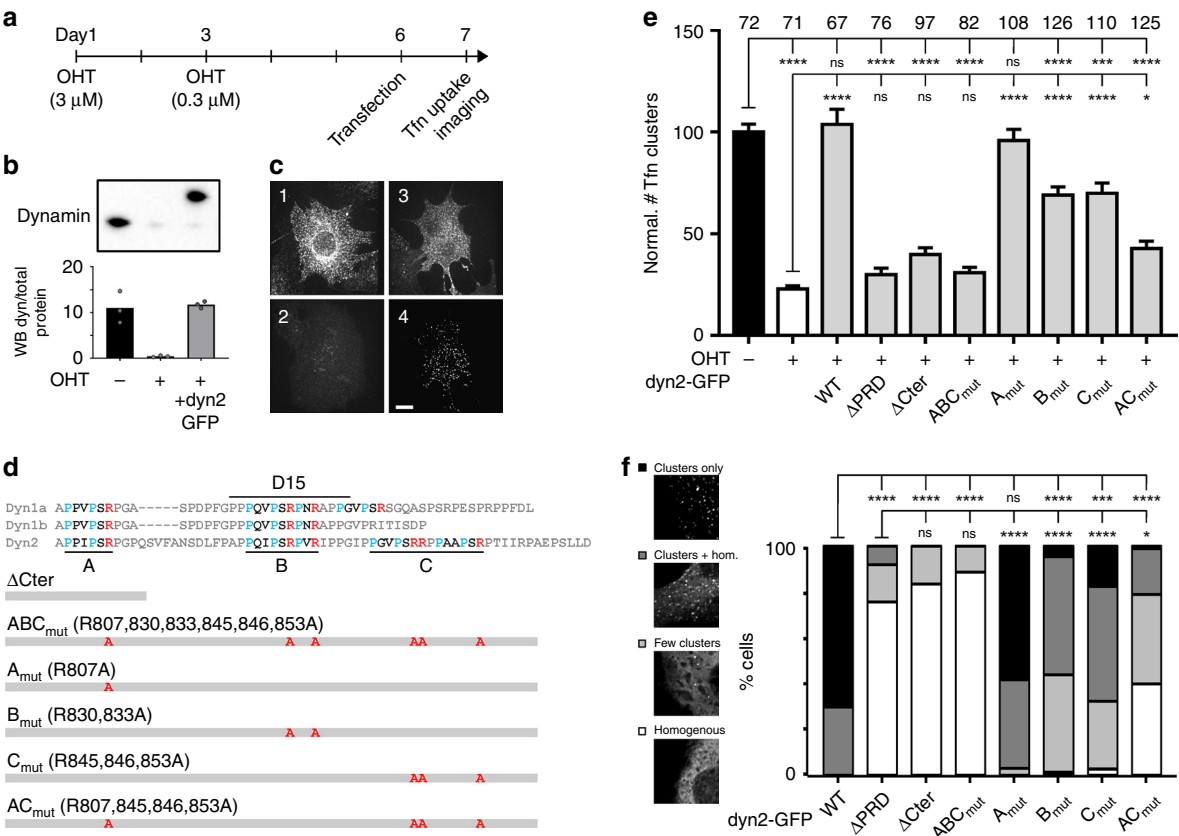

**Fig. 1** Rescue of endocytosis in dynamin TKO cells with PRD mutant dynamins. **a** Protocol for dynamin knock-out and rescue. After 6 days of gene excision of the three dynamin genes following hydroxytamoxifen (OHT) treatment, cells are transfected with electroporation. Tfn uptake and imaging assays are performed the next day. **b** Western blot with pan-dynamin antibody on untreated cells, cells treated with OHT or cells treated with OHT and transfected with dynamin2-GFP. Note the migration at higher molecular weight of dyn2-GFP. Bottom, quantification in three experiments of the WB signal normalized to the total amount of protein. See Supplementary Fig. 1A for full gels and blots. **c** Example images (red channel) of cells incubated with Tfn-A458 for five minutes, washed and fixed. (1) Not treated with OHT, (2) treated with OHT, (3) treated with OHT and transfected with dyn2-GFP. (4) GFP channel of the cell shown in (3). Scale bar 10 μm. **d** Sequences of the C terminal parts of the PRD of dyn1 (a and b splice variants) and dyn2, with the class-II SH3 binding motifs A, B and C indicated. Below, mutants of dyn2 used to rescue Tfn-A568 uptake with the corresponding mutations. **e** Quantification of Tfn-A568 uptake, expressed as the density of detected clusters for cells treated with OHT and transfected with dyn2-GFP mutants as indicated. The number of cells imaged in four different experiments is indicated on top of each condition. The graphs show the average ± SEM for each dyn2 mutant. The adjusted $p$ values of one-way ANOVA followed by Tukey's multiple comparison tests are shown in Supplementary Table 3. **f** Quantification of dyn2-GFP localization in transfected TKO cells. Each cell was evaluated blind with a score ranging from homogenous labelling (white) to punctuate labelling without homogenous (black), with intermediates (mostly homogenous with few clusters, light grey; some homogenous with distinct clusters, dark grey). Examples in the left illustrate this scoring. Histograms show the proportion of each category of labelling. Stars indicate statistical significance (Kruskal–Wallis test followed by Dunn's multiple comparison tests, p values in Supplementary Table 4)

living cells. We compared the recruitment kinetics of dyn2-GFP mutants in TKO cells with that of genome-edited SKMEL cells expressing dyn2-GFP in the endogenous gene locus[8]. For this analysis, we chose four different dyn2-GFP constructs that showed various degrees of CME rescue: WT, ΔCter, B$_{mut}$ and AC$_{mut}$ (Fig. 2a). As observed above in fixed cells, mutated dyn2-GFP appeared as a combination of clustered and homogenous fluorescence (Fig. 2a). The average cell fluorescence outside clusters was thus significantly higher in mutants compared to the WT (Fig. 2b). These clusters were transient and the frequency at which these clusters could be detected correlated with the ability of the constructs to rescue CME: highest for dyn2-GFP-WT re-expression, intermediate for dyn2-GFP-B$_{mut}$, low for dyn2-GFP-AC$_{mut}$ and essentially null for dyn2-GFP-ΔCter where fluorescence was completely homogenous (Fig. 2c). Interestingly, the peak amplitude (Fig. 2d) and kinetics (Fig. 2e) of these recruitment events were similar in all conditions, including in genome-edited cells. We verified that the higher frequency of transient dynamin

clusters detected in the latter (0.34 ± 0.04 ev min$^{-1}$ μm$^{-2}$, $n = 15$ SKMEL vs 0.20 ± 0.02 ev min$^{-1}$ μm$^{-2}$, $n = 9$, MEF; $p = 0.006$) was indeed due to the difference in cell types rather than an artefact of the position of the GFP tag by transfecting TKO cells with wild-type dyn2 tagged at three different positions. This did not affect the frequency, amplitude and kinetics of dynamin recruitment events (Supplementary Fig. 2).

Considering that amphiphysins are major interactors of dynamins and are essential for CME[17,18], their recruitment to the plasma membrane could be affected by the dynamin mutants. We co-expressed amphiphysin2-mCherry with dyn2-GFP mutants in TKO cells. In all cases, amphiphysin2-mCherry appeared punctate and, except in the case of the homogenous ΔCter mutant, partly colocalized with dyn2-GFP (Fig. 2f). Similar to dynamin, amphiphysin2-mCherry also displayed transient clusters. Their frequency or amplitude were not affect by the co-expression of any of the dyn2-GFP mutant construct (Fig. 2g, h). Therefore, amphiphysin recruitment to CCPs does not depend on

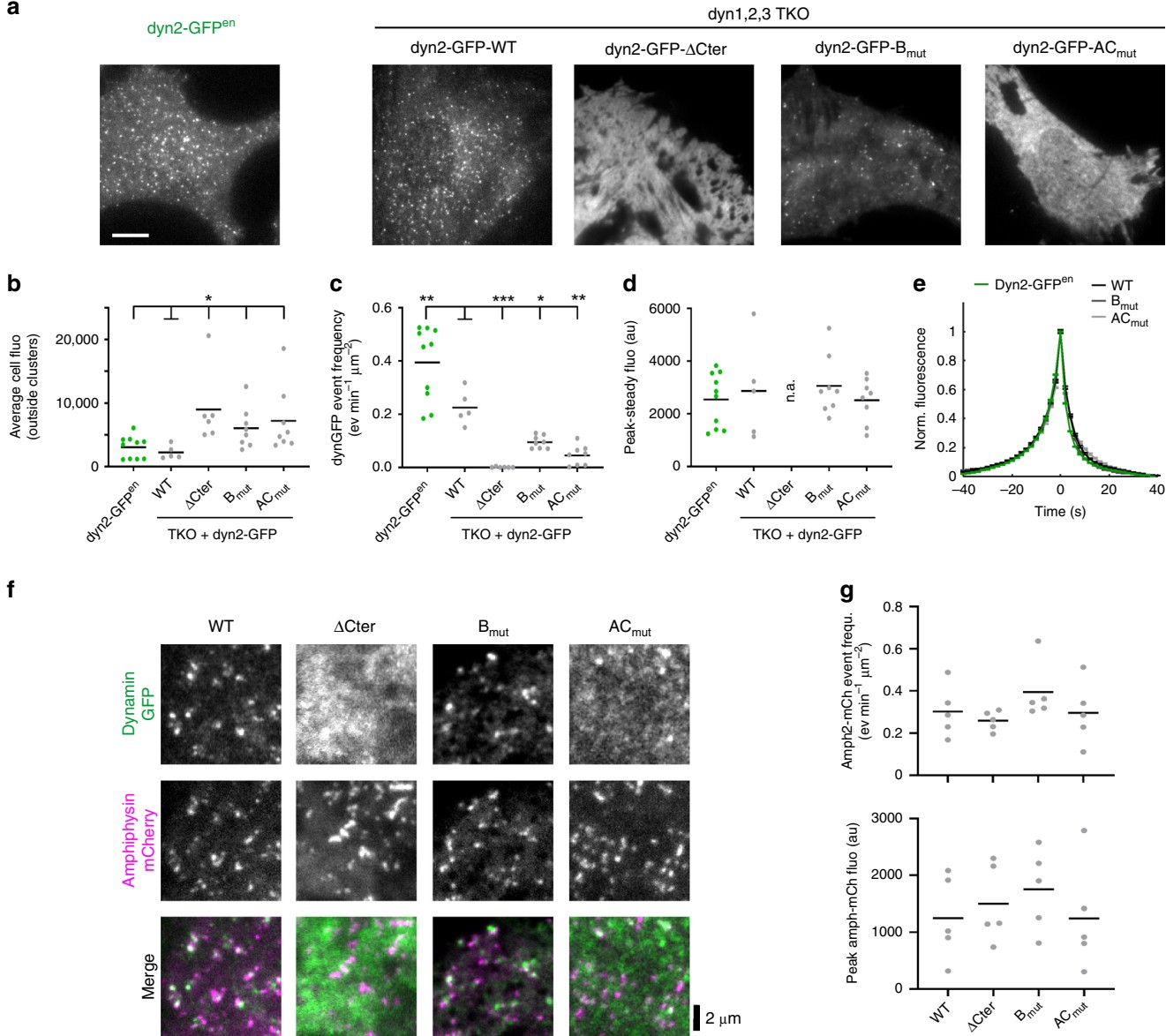

**Fig. 2** Frequency and kinetics of dyn2 recruitment events in dynPRD mutants. **a** TIRF images of living SKMEL cells expressing dyn2-GFP in both endogenous loci (left) or dynamin TKO MEF cells transfected with the indicated dyn2-GFP mutants (right). Scale bar 10 μm. **b** Average fluorescence outside the dyn2-GFP clusters for the various dyn2-GFP mutants. The number of cells recorded is indicated at the top of the graph. **c** dyn2-GFP recruitment event frequencies for the same cells as in (**b**). **d** Average amplitudes of peak dyn2-GFP events for the same cells as in (**b**), except for cells transfected with dyn2-GFP-ΔCter (n.a.). **e** Normalized average fluorescence of all the events detected, aligned to their peak fluorescence. The total number of events averaged was 36070 for dyn2-GFPen cells and 18678, 127, 5916 and 6116 for TKO cells transfected with dyn2-GFP-WT, dyn2-GFP-ΔCter, dyn2-GFP-B_{mut} and dyn2-GFP-AC_{mut}, respectively. **f** TIRF images of portions of dynamin TKO cells co-transfected with amphiphysin-mCherry and the indicated dyn2-GFP mutants. Scale bar 2 μm. **g** Frequencies (top) and peak amplitudes (bottom) of amph-mCherry recruitment events recorded in dynamin TKO cells co-transfected with the indicated dyn2-GFP mutants. N = 5–7 cells for each condition, no significant difference with WT in any condition

its interaction with dynPRD. Taken together, these results suggest that the PRD-SH3 interaction regulates the probability of recruiting dynamin but that once it is initiated, the process unfolds in a stereotyped manner to complete vesicle formation.

Next, we directly assessed membrane scission achieved by the dyn2 mutants using the ppH assay. This assay relies on the overexpression of TfR fused to pH sensitive supereccliptic pHluorin (TfR-SEP) as an endocytosis marker to detect the formation of single CCVs in living cells with a temporal resolution of 2 s[4,26,28]. We co-transfected TKO cells with TfR-SEP and dyn2-mCherry constructs. TfR-SEP was clustered at clathrin coated structures (CCSs) in cells co-transfected with dyn2-mCherry-WT or dyn2-

mCherry-B_{mut} (Fig. 3a), but was homogenously distributed on the plasma membrane in cells expressing dyn2-mCherry-ΔCter or dyn2-mCherry-AC_{mut} (Fig. 3b). This is likely due to the redistribution of TfR to the plasma membrane and saturation of CCSs as already observed in TKO cells[6]. We detected nascent CCVs with the ppH assay in cells co-transfected with dyn2-mCherry and dyn2-mCherry-B_{mut} at frequencies of $0.025 \pm 0.009$ and $0.004 \pm 0.002$ event min$^{-1}$ μm$^{-2}$, respectively. This significant difference ($p = 0.016$) reflects the changes in Tfn uptake observed (Fig. 1e). On the other hand, we observed some moving vesicles at pH 5.5 in cells co-transfected with dyn2-mCherry-ΔCter or dyn2-mCherry-AC_{mut} but the high intracellular background and lack of clustering

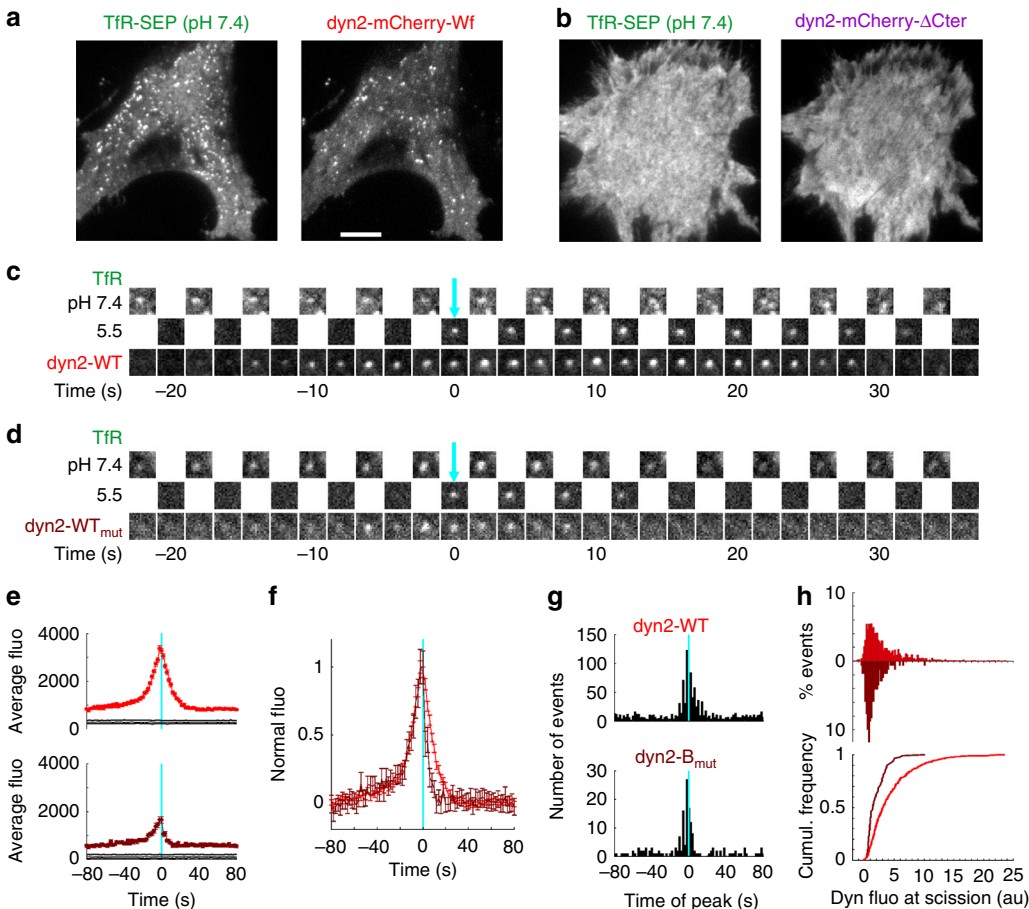

**Fig. 3** Kinetics of dynamin recruitment to forming CCVs measure with ppH. **a**, **b** Images of a TKO cell co-transfected with TfR-SEP and dyn2-mCherry-WT (**a**) or dyn2-mCherry-ΔCter (**b**). Note the punctuated pattern of both markers with dyn2-WT which becomes homogenous with dyn2ΔCter. Scale bar 10 μm. **c**, Example of a scission event recorded in the cell shown in A. Note the transient recruitment of dyn2-mCherry-WT which peaks around time 0. **d** Same as **h** for a TKO cell co-transfected with TfR-SEP and dyn2-mCherry-B$_{mut}$. **e** Average fluorescence of dyn2-mCherry of events recorded in cells co-transfected with TfR-SEP and dyn2-mCherry-WT (top, red trace, 1193 events in 5 cells) or dyn-mcherry-Bmut (bottom, brown trace, 171 events in 7 cells) and aligned to the time of scission (cyan line). The black lines indicate 95% confidence intervals for significant recruitment. **f** The curves in (**e**) are normalized to their peak. **g** Histograms of the timing of peak dyn2 recruitment in single events relative to scission (cyan lane). **h** Histograms (top) and cumulative distributions (bottom) of the amplitude of dyn2-mCherry at the time of scission for events recorded with a peak recruitment within 10 s before or after scission (580 and 97 events in cells transfected with dyn2-mCherry-WT, in red, and dyn2-mCherry-B$_{mut}$, in brown, histogram reversed, respectively)

on the plasma membrane precluded further characterization. We then moved on to compare endocytic events in cells transfected with the endocytically competent mutants. Dyn2-mCherry-WT was readily recruited in a burst of fluorescence that peaked at the time of CCV formation (Fig. 3c–f) as observed previously[4,28]. Dyn2-mCherry-B$_{mut}$ recruitment looked comparable but, remarkably, the peak fluorescence at the time of scission was smaller (Fig. 3e). We hypothesised this could be due to a desynchronization of dynamin recruitment relative to scission. However, the recruitment kinetics of both dynamin mutants relative to CCV formation (Fig. 3f), as well as the histograms of peak recruitment of individual events (Fig. 3g), were similar before scission, arguing against this possibility. On the other hand, the fluorescence of dynamin at the time of scission for individual events was smaller with a narrower distribution for dyn2-mCherry-B$_{mut}$ than for dyn2-mCherry-WT (Fig. 3h). Therefore, we conclude that destabilisation of the dynPRD-SH3 interaction reduces the amount of dynamin recruited at the time of scission. This observation goes in line with the reduced probability of mutant dynamin being recruited by amphiphysin as suggested above.

**Binding of divalent D15-based ligands for amphSH3 dimers.** To further test the model that multiple SH3 binding motifs in dynPRD are necessary for its recruitment to CCSs, we designed synthetic peptide-based ligands that preserved the key binding properties of dynamin in its capacity to interact simultaneously with multiple SH3 domains. We anticipated that such multivalent ligands should inhibit CME with higher efficiency than the classical, monovalent, D15 peptide. First, we synthesized peptides derived from dyn1-PRD (named D44 thereafter) and dyn2-PRD (D54-dyn2) (Fig. 4a) and compared their binding properties to recombinant amphSH3 with that of the D15 peptide by enzyme-linked immunosorbent assay (ELISA). Long peptides bound in a similar manner (K$_D$ 1.0 and 1.1 μM for D44 and D54-dyn2, respectively) but more strongly than D15 (Fig. 4b). Moreover, mutating motif A in D44 abolished high affinity binding (D44-R11A in Fig. 4b), consistent with the hypothesis that several class II binding motifs stabilize the interaction with several amphSH3 immobilized at a high density on the substrate. To analyse the effect of the multimerisation of SH3 domains in greater detail, we produced them with controlled valences: (i) monomeric as a

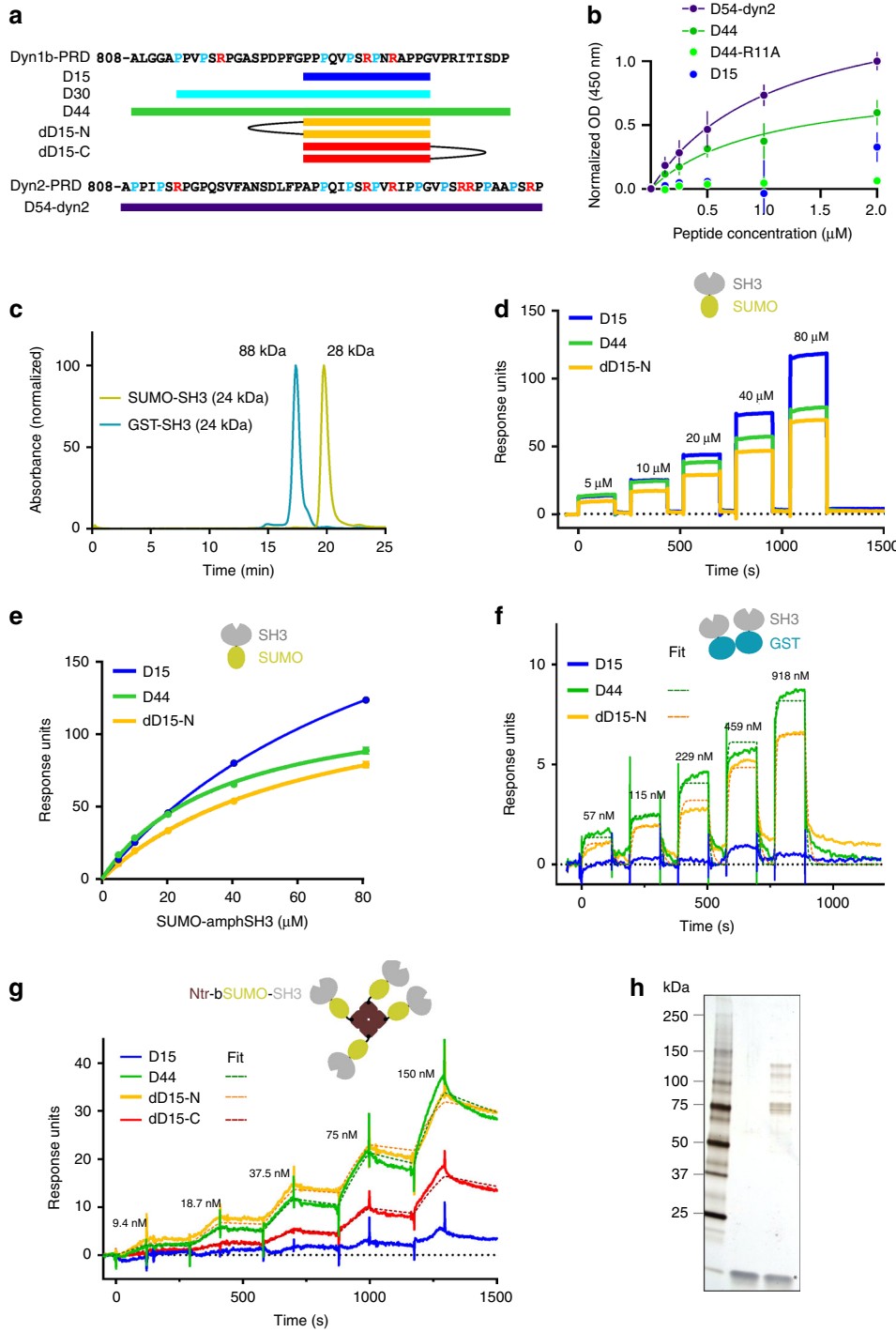

**Fig. 4** Binding of mono- and multimeric SH3 domains to D15-based peptides. **a** Scheme of the peptides used in this study, based on dynamin1b (top) and dynamin2 (bottom) PRDs. **b** ELISA to evaluate the respective binding properties of biotinylated peptides to surface-bound SUMO-SH3 domains ($K_D = 1.12$ and $1.01\,\mu M$ for D54-dyn2 and D44 respectively, the other peptides were not fitted). Each data point represents the average of two independent experiments with two technical replicates each. **c** Retention times of SUMO-SH3 and GST-SH3 measured by size exclusion chromatography. The theoretical molecular weights are indicated. For SUMO-SH3 the elution peak corresponds to 28 kDa, close to the expected monomer, whereas for GST-SH3 it corresponds to 88 kDa, close to the expected size of a dimer. **d** Representative sensorgrams of SUMO-SH3 binding at indicated concentrations to immobilized peptides. Note the fast on and off rates for all peptides consistent with weak and transient interactions. **e** Steady-state $K_D$s calculated from the data in (**d**) (three replicate experiments): $102 \pm 2\,\mu M$ (D15), $37 \pm 2\,\mu M$ (D44), $66 \pm 3\,\mu M$ (dD15-N). **f** Representative sensorgrams of GST-SH3 binding at indicated concentrations to immobilized peptides. The scheme on top (GST in blue, SH3 in grey) indicates that this protein likely forms dimers (see **b**). Dotted lines represent fits with estimated $K_D$s of 572 nM (D44) and 509 nM (dD15-N). **g** Representative sensorgrams of Ntr-bSUMO-SH3 binding at indicated concentrations to immobilized peptides. The scheme on top shows the multimeric complex formed by bSUMO-SH3 bound to tetrameric neutravidin (brown). Dotted lines represent fits with estimated $K_D$s of 17 nM (D44), 4.5 nM (dD15-N) and 34 nM (dD15-C). Note the change in kinetics (slower off rates) for D44 and the multimeric peptides (dD15-N and dD15-C). **h** Silver stain of rat brain lysate pull-down with no peptide (middle) or biotinylated dD15-N (right)

fusion to the small ubiquitin-like modifier (SUMO-amphSH3), (ii) dimeric[29] fused to the widely used glutathione S-transferase C terminal domain (GST-amphSH3)[14,18,20,30,31] as confirmed by size exclusion chromatography (Fig. 4c), or (iii) tetrameric by reacting a biotinylated SUMO-amphSH3 with tetrameric neutravidin (Ntr, leading to a Ntr-bSUMO-amphSH3 complex). Using surface plasmon resonance (SPR), we first measured the dissociation constants of mono- and divalent dyn1-derived peptides D15 and D44 with SUMO-amphSH3. In both cases, the affinities were weak with $K_D$s in the high micromolar range (37 μM vs. 102 μM respectively, Fig. 4d, e) and the binding kinetics were too fast to be accurately measured. In contrast, robust binding of the dimeric GST-amphSH3 to D44 was observed, with an estimated $K_D$ of 572 nM, while there was no measureable binding to D15 below 1 μM (Fig. 4f). We attribute this dramatic gain in affinity to the fact that dimeric GST-amphSH3 can simultaneously bind to the two SH3 binding motifs present in D44, thereby stabilizing the interaction. Finally, the tetrameric Ntr-bSUMO-amphSH3 complex showed an even higher affinity for D44 (17 nM) while no binding was observed for D15 in the tested concentration range (Fig. 4g). Of note, these increases in affinity for the dimeric and tetrameric amphSH3 assemblies were accompanied by slower dissociation rate constants, a hallmark of multivalent interactions.

Second, we designed synthetic ligands that mimic the presence of multiple SH3 binding motifs in dyn1-PRD. These divalent peptide variants consist of two D15 peptide motifs linked through their N or C termini by PEG linkers (dD15-N and dD15-C: Fig. 4a, Supplementary Fig. 3 and Supplementary Table 1). They behaved in a qualitatively similar manner as D44 such that they bound dimeric GST-amphSH3 and tetrameric Ntr-bSUMO-amphSH3 much more efficiently than they did monomeric SUMO-amphSH3. dD15-N bound SUMO-amphSH3, GST-amphSH3 and Ntr-bSUMO-amphSH3 with $K_D$s of 65 μM, 509 nM and 4.5 nM, respectively, while dD15-C bound Ntr-bSUMO-amphSH3 with 34 nM (Fig. 4c–g). To further characterize the functional relevance of the increased valence of those peptides, we performed affinity isolation assays on brain lysates. In our conditions, D15 was unable to pull down any detectable amount of protein whereas dD15-N isolated a number of proteins (Fig. 4h). Mass spectroscopy analysis of the pull-down material indicated that while D15 failed to show any significant enrichment in comparison to control conditions (absence of peptide), the material the co-precipitated proteins by dD15-N was highly enriched in CME related proteins and in particular in known interactors of dynamin such as SH3kbp1 (known as CIN85 in human)[32], intersectin1-2, amphiphysin1 and amphiphysin2 (also known as BIN1), endophilinA1-2 and SNX18 (Supplementary Table 2). Notably, all these interactors either contain multiple SH3 domains which bind dynamin (CIN85, intersectins) or a BAR domain which enables dimer formation (endophilins, amphiphysins, SNX18)[33]. Our in vitro data therefore support the idea that multimerisation of SH3 domains together with the presence of multiple SH3 binding motifs in dynPRD play an essential role in increasing the affinity of their interaction by avidity effects.

**Divalent D15-based peptides inhibit CME in living cells**. We then developed an assay to determine whether D44 (bearing motifs A and B) and divalent D15 peptides exhibited an increased inhibitory activity over the monovalent D15 on CME. To do so, we monitored cellular endocytic activity using the ppH assay with new developments for fully automated analysis on a large number of cells (See "Methods" and Supplementary

Fig. 4). We combined this assay, which monitors in real time the endocytic activity of cells, with the patch-clamp technique in order to dialyse the relevant peptide at a known concentration inside the cellular cytoplasm to inhibit endocytosis acutely. Each recording consisted of 5 min in cell-attached mode (CA) with no dialysis followed by 10 min in whole cell mode (WC) during which the inhibitors were dialysed (Fig. 5a, b). Each recording therefore had its own internal control condition, i.e., the cell-attached mode, from which we obtained the frequency ratio $f = F_{WC}/F_{CA}$, where $F$ is the event frequency recorded during the indicated mode. Of note, the electrical parameters of patch clamp recordings were similar in all conditions (Supplementary Fig. 5). To assess the innocuousness of this assay, we monitored the endocytic activity of unperturbed cells vs. cells dialysed with a control solution (see methods for composition). Patching the cells minimally affected their endocytic activity over a 10 min period (event frequency $f = 67.3 \pm 4.3\%$, $n = 60$, vs. $90.7 \pm 14.9\%$ in no patching recordings, $n = 8$, $p = 0.58$).

To assess the proper dialysis of the inhibitors in this assay, we first dialysed the cells with a solution in which GTP was replaced by its non-hydrolysable form, GTPγS, which prevents the GTP-hydrolysis dependent activity of dynamin[22,24]. As expected, this rapidly abolished CCV formation (Fig. 5c, d) ($f = 11.2 \pm 4.1\%$, $n = 4$, $p < 0.0001$). We then compared the effect of the D15 peptide when patched inside the cells with its effect when bath applied in its myristoylated, cell-permeable form (myrD15[34,35]). D15 dialysed at 1 mM with a patch pipette partially inhibited endocytosis within 10 min ($f = 38.2 \pm 3.9\%$, $n = 22$, $p = 0.0002$, Fig. 5b–d). Similarly, myrD15 (10 μM) incubated with cells for 20 min, partially inhibited endocytosis (Supplementary Fig. 6A, B). The corresponding non-sense, inactive sequences had no effect on endocytosis (Fig. 5d and Supplementary Fig. 6B). We confirmed that inhibitor concentration could be controlled in the patch-clamp assay by varying the concentration in the pipette. When used at 100 μM, D15 indeed hardly had any effect on endocytosis ($f = 59.0 \pm 5.0\%$, $n = 13$, $p = 0.98$) (Fig. 5d), which is consistent with its low affinity for its target amphiphysin. Of note, we observed that myrD15 also partially inhibited the endocytosis of the β2 adrenergic receptor (β2AR) (Supplementary Fig. 6C, D), another cargo of CME[36] but did not significantly affect the internalisation of the closely related β1 adrenergic receptor (β1AR) (Supplementary Fig. 6E, F), which uses a dynamin mediated, clathrin independent pathway[37].

We thus moved on to compare the inhibitory activities of multivalent dynPRD derived peptides on CME. The dialysis of 1 mM of D44 with a patch pipette abolished endocytosis almost completely, ($f = 20.6 \pm 5.2\%$, $n = 15$, $p < 0.0001$). Even 100 μM of either D44 or D30, a trimmed down version of D44 bearing the two SH3-binding motifs (Fig. 4a), partially inhibited endocytosis ($f = 41.4 \pm 9.6\%$ and $41.7 \pm 5.4\%$, $p = 0.03$ and 0.04, respectively). Finally, the divalent peptides dD15-N and dD15-C also had a clearly increased efficiency over D15 (Fig. 5b–d), with the C terminal version being the most potent D15-derived inhibitor of endocytosis ($f = 13.3 \pm 3.7\%$, $n = 6$, $p < 0.0001$). The effects of the various conditions on CME measured with the ppH assay were consistent with the changes in cell capacitance, measured with patch-clamp in whole cell mode, which reflect the balance between exocytosis and endocytosis (Supplementary Fig. 5F). Overall, the increased efficiency of D44 and divalent D15 peptides in inhibiting endocytosis in living cells is reminiscent of the in vitro behaviour described above. Thus, these data support a model in which dynamin binding partners are present in a multimeric configuration in the cellular context for efficient recruitment of dynamin at endocytosis sites.

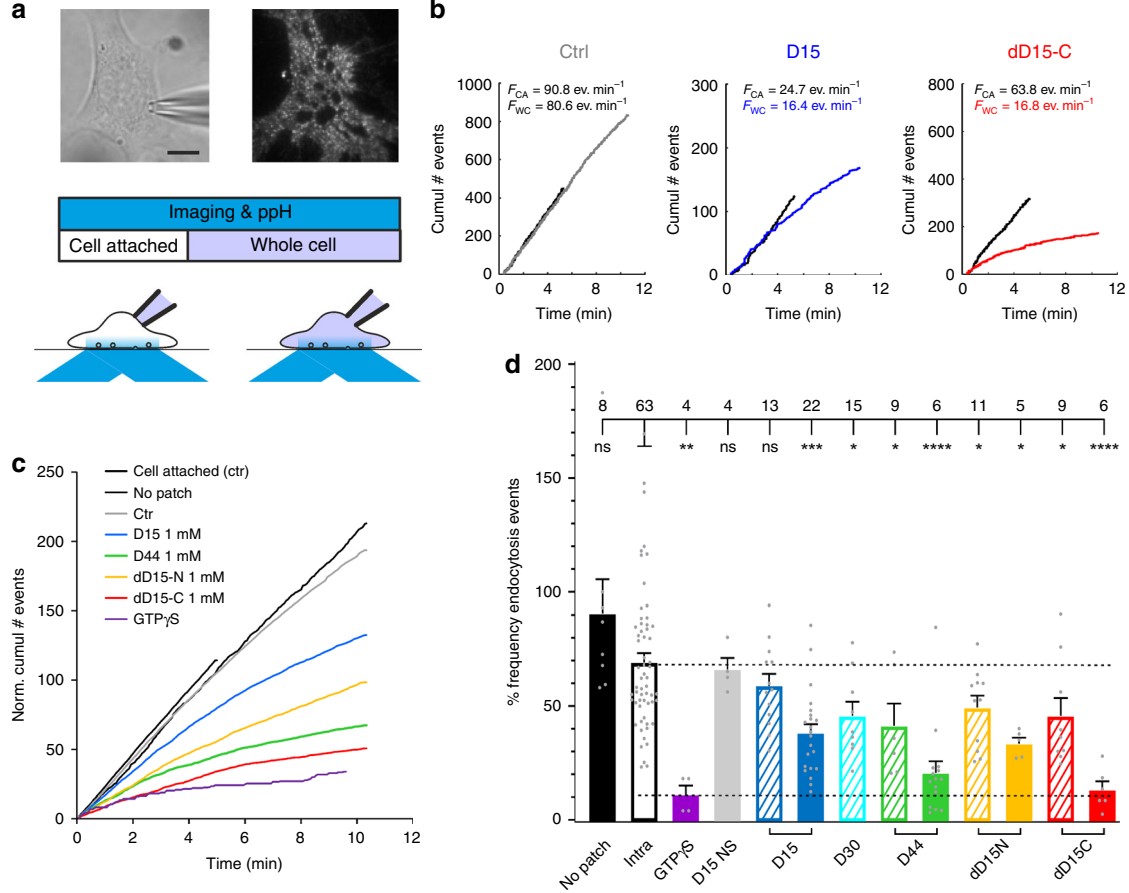

**Fig. 5** Effect of D15-based peptides on CME monitored with the ppH assay. **a** Experimental design. Top, images of a 3T3 cell recorded with a patch-clamp electrode (seen on the transmitted light image, left) and transfected with TfR-SEP imaged with TIRF microscopy (right). The cells are recorded for at least 15 min with the ppH assay. In the first 5 min, the cell is in the cell attached configuration (left) and in the last 10 min in the whole cell configuration (right). Scale bar 5 μm. **b** Examples of recordings of cells with internal solutions containing no peptide (Ctrl, left), 1 mM D15 (middle) or 1 mM dD15-C (right). Black lines represent the cumulative number of detected endocytosis events over time in cell attached configuration. They are all straight lines, indicating that the event frequency remains constant during recording. This frequency $F_{CA}$ is written on top of the graph. Lines in colour represent the cumulative number of events in whole cell configuration. Note that at early time points these curves are tangential to the corresponding curves in cell attached, indicating the same endocytosis activity. In the example with the control solution, the slope of this curve remains constant with only a slight deflection towards the end of the recording. On the other hand, the deflections are much more marked with solution containing inhibitory peptides. The frequency $F_{WC}$ in the last two minutes of the whole cell recording is indicated on top of the graphs. **c** Averages of curves normalized to the corresponding cell attached recordings and displayed as in (b) for all the conditions tested in this study. **d** Average ± SEM of the ratio of event frequency measured 8–10 min in whole cell over event frequency 3–5 min in cell attached for the same cell. The number of recordings for each condition is indicated on top of the graph, and each individual measure is indicated by a grey circle. Recording solutions contain 100 μM (hatched bars) or 1 mM (plain bars) of the indicated peptide. Stars indicate statistical significance for difference (1-way ANOVA followed by Tukey's multiple comparison tests, Supplementary Table 5)

## Discussion

Taken together, the data presented here support the idea that dynamin-SH3 interactions rely on the presence of multiple interaction sites to achieve successful membrane scission.

We show that at least two class-II consensus SH3 interaction motifs have to be present in the C-terminal part of dynPRD for mutant dynamin to regulate dynamin recruitment and rescue the endocytic defects observed in dynamin TKO cells. The central B motif, from which the D15 is derived, is important but not essential; the C-terminal C motif appears to play a similar role. Interestingly, this C motif is only found in dynamin 2, which could explain its better ability to rescue CME in cell lines than the neuronal dynamin 1[27] and the greater sensitivity of neuronal cells to D15 cytoplasmic dialysis[24]. Similar to motif B (PSRPVRI), motif C also contains an extra arginine after the class II binding motif (PGVPSRR), making it a preferred ligand for amphSH3[14,38]. However, we clearly show with TKO cell rescue

experiments that isolated motifs cannot stably bind multivalent targets. We propose that adjacent SH3 binding motifs on dynPRD stabilize its binding to neighbouring multiple SH3 domains. One motif (B or C) shows specificity for amphSH3 whereas the stabilizing motif may or may not show any specificity, but the proximity of binding sites is enough to stabilize the interaction by avidity effects.

Our second set of experiments with competing peptides is entirely consistent with this model. We showed that D15 alone was a poor binder to amphSH3 in either its mono- (SUMO-amphSH3) or oligomeric (GST- or Ntr-bSUMO-amphSH3) forms, whereas both the elongated D44 peptide and the divalent D15 peptides bound with higher affinity to oligomeric amphSH3. Alternatively, D44 could bind amphSH3 with higher affinity because of additional motifs binding to other, non-canonical sites on a single SH3 domain, as shown for dynPRD binding to syndapin SH3 domain[30]. However, it is unlikely to be the case here

because the differential effect we observe takes place only with oligomeric SH3 constructs. Our in cellulo data were also consistent with these properties. Indeed, dialysing D44 or divalent D15 peptides inside 3T3 cells fully inhibited CCV formation whereas inhibition by monovalent D15 was only partial. To our knowledge, despite the recognized importance of multivalent interactions for the endocytic process[5], divalent peptides have never been used to interfere with endocytosis in living cells. Considering the moderate inhibitory activity of D15, the D44 and dD15 peptides described here appear to be better tools for this purpose. However, long peptides such as D44 are more difficult to obtain than shorter peptides such as D15 due to the inherent limitations of solid-phase peptide synthesis. In this context, divalent peptides present the advantage of being accessible with significantly less synthetic steps as compared to D44[39] (see "Methods" for details). dD15-N appeared to bind slightly more strongly to tetrameric amphSH3 than dD15-C (4.5 nM vs 34 nM). However, dD15-C was more potent in inhibiting CME in living cells than dD15-N, suggesting that the precise arrangement and dynamics of SH3 domains recruiting dynamin to nascent endocytic vesicles cannot be easily reproduced in vitro. In all cases, these tools will help to decipher the role of multimeric interactions for dynamic protein assemblies in cells, an emerging field in cell biology[40].

Our results shed light on the mechanisms of dynamin recruitment to CCSs (Fig. 6). Efficient recruitment of dynamin requires that multiple adjacent class II motifs on dynPRD interact simultaneously with multiple SH3 domains. Consequently, these multiple SH3 domains must be brought together, either in a single protein or through protein oligomerization. In vitro, dynPRD interacts with many SH3 domain containing proteins[9] which display distinct recruitment patterns relative to CCV formation[4]. Dynamin is recruited early at low levels in the formation of CCSs[4,7] mainly at the edge of the clathrin lattice[41]. Interestingly, intersectin has five SH3 domains, of which three can bind dynamin[42] and can thus interact with multiple motifs on dynPRD regardless of its oligomerization state. Moreover, intersectin is among the earliest proteins recruited to CCSs[43] and is also

located at the edge of clathrin lattices[41], making it a prime candidate for an early interactor of dynamin. At later stages, dynamin accumulates rapidly to the neck of forming vesicles at the same time as endophilins, amphiphysins, and SNX9/18[4]. All these proteins contain a single SH3 domain as well as a BAR domain. BAR domains form dimers that can bind to or induce curved membranes[33]. Among those, amphiphysins are the most likely to be critical for dynamin function in CME because dialysis of amphSH3 has strong effects on CCV formation and transferrin uptake[18–20] and knock-down of amphiphysins has the strongest effects of these proteins on CME[17]. Interestingly, we found that myrD15 partially blocks CME of TfR and β2AR but has no significant effect on the internalization of β1AR, which depends on dynamin and endophilin but not on clathrin and amphiphysin[37,44]. D15-derived peptides could thus antagonize specific forms of dynamin dependent endocytosis such as CME which critically depend on the interaction between dynamin and amphiphysin. We propose that as long as dynamin and amphiphysin are found in the cytoplasm as monomers or perhaps homo-dimers[33,45], they cannot strongly bind to each other due to the low affinity of their interaction and because the two SH3 domains are too far apart. They would just form short-lived, transient complexes. However, when amphiphysin is recruited to the neck of vesicles thanks to its curvature sensing properties, its SH3 domains may rearrange in such a way that they would be in close proximity and made available for dynamin binding[46]. Whether amphSH3 domains are optimally placed to face dynPRD in a cellular environment remains unknown. The structure of dynamin has been solved by crystallography[47,48] or cryo electron microscopy[49–51] in many different conformations but always without its unstructured PRD. Moreover, dynPRD binding to amphSH3 competes with an intramolecular interaction between its BAR and SH3 domains as was seen for other proteins involved in CME such as syndapin 1[52] or endophilin A1[46]. Therefore, a reciprocal stabilisation of the interaction between dynamin and amphiphysin could occur through oligomerisation. The open configuration of amphiphysin would only be possible once the neck of the vesicle is tight enough for amphiphysin to form a

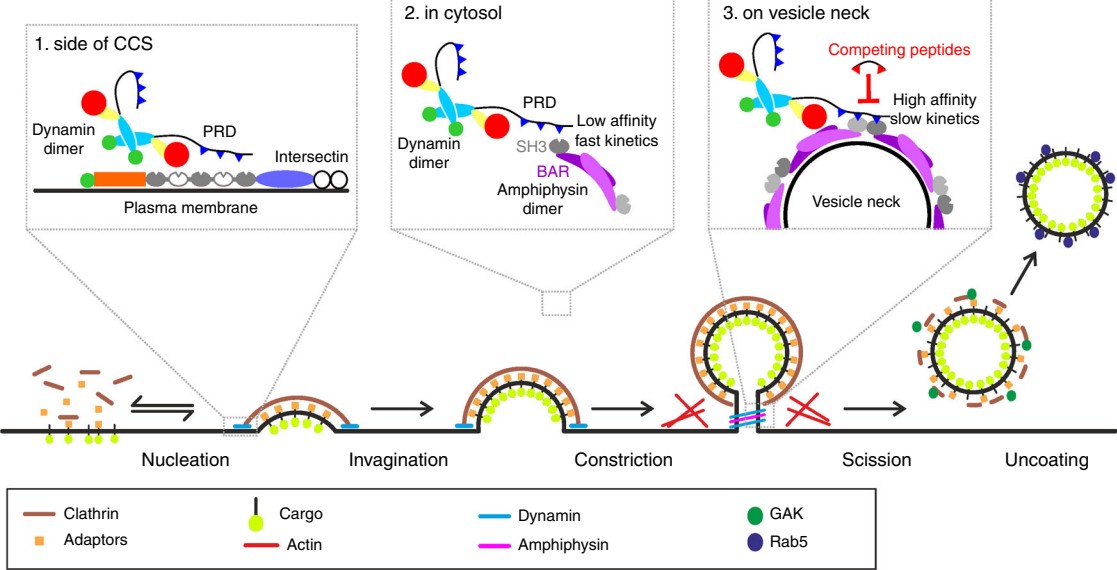

**Fig. 6** Model of dynamin recruitment to forming CCVs. Model of dynamin recruitment by its PRD domain with SH3 containing proteins. (1) At early stages of CCV formation dynamin interacts with intersectin which has multiple dynamin interacting (grey) and non-interacting (white) SH3 domains. (2) In the cytosol, dynamin forms a dimer, as well as SH3 domain containing proteins such as amphiphysin. The SH3 domains of the dimer are too far apart to stabilize the interaction with the PRD via multiple binding motifs. (3) On the vesicle neck, two dimers of amphiphysin are arranged such that two SH3 domains are next to each other. This ensures high affinity of dynamin with this configuration enabling the fast recruitment of dynamin

scaffold around it. Dynamin would bind this multimeric scaffold, stabilising amphiphysin's new conformation. This would provide a seed for further recruitment of both proteins around the vesicle neck in a positive feedback mechanism. Consistent with this model is the observation that the start of dynamin burst recruitment ~20 s before scission coincides with that of amphiphysin and endophilin[4]. Also, affecting dynamin recruitment kinetics in turn affects endophilin recruitment kinetics[53]. The coincidence of this precise geometry and the advent of avidity conferred by multimerisation as highlighted by this study therefore provide an elegant explanation for the timely accumulation of dynamin at the moment of membrane scission.

## Methods

**Peptide synthesis**. All the peptides used in this study were synthetized according to the procedure below, except for D54-dyn2, D44(R11A) which were purchased from Covalab, and myrD15 and myrCTL from Tocris.

**Chemicals for peptide synthesis**. 9-fluorenylmethoxy-carbonyl (Fmoc)-protected amino-acids were from GenScript USA Inc. 2-(1H-Benzotriazol-1-yl)-1,1,3,3-tetramethyluronoium hexafluorphosphate (HBTU), [N1-(Fmoc)-1,13-diamino-4,7,10-trioxatridecan-succinamic acid (Fmoc-TTDS) and Fmoc-Lys($N_3$)-OH were from Iris Biotech GmbH. 1-hydroxy-benzotriazole (HOBt), N,N'-diisopropylethylamine (DIPEA), N-methylpiperidine, acetic anhydride, pyridine, biotin, 4-pentynoic acid, Triisopropylsilane (TIPS), TRIZMA base and 2,5-Dihydroxybenzoic acid (DHB) were from Sigma-Aldrich. Trifluoroacetic acid (TFA), acetonitrile (MeCN), dichloromethane (DCM), N,N-dimethylformamide (DMF) and diethyl ether were from Fisher scientific. Fmoc-Dab(Alloc)-OH was from Bachem AG. N-methyl-2-pyrrolidone (NMP) was from Applied Biosystems.

**Procedure for peptide synthesis**. Monomeric peptides were synthesised at 0.05 mmol scale on Fmoc-NOVA PEG rink amide AM LL resin (Novabiochem) and dimeric peptides on PAL NovaSyn TG resin (Novabiochem). Amino-acids were assembled by automated SPPS on a CEM micro-waves Liberty-1 synthesiser (Saclay, France) except for C-terminally linked dimeric peptides. Coupling of PEG motifs and N-terminal derivatisations were also performed manually. For automated synthesis, Fmoc protecting groups were cleaved with 4-methylpiperidine in DMF (1:5 v/v) containing 0.1 M HOBt using a combination of short (35 W, 75 °C, 30 s) and a long cycle (35 W, 70 °C, 180 s). Couplings of Fmoc-protected amino acids (0.2 M) were carried out in the presence of HBTU (0.25 M) and DIPEA (1 M) using standard coupling cycles (20 W, 70 °C, 300 s, except for Fmoc-Arg(Pbf)-OH (0 W, RT, 1500 s followed by 20 W, 70 °C, 300 s). For the synthesis of D44, we evaluated a priori difficult couplings according to[54]. We thus introduced a systematic double coupling on Y1, A2, L3, G5 (N-terminus amino acids) V37, P38, I40 and T41 (bulky amino acids). Other double couplings were then introduced empirically at positions P7, S11, S16, P22, P23, 33, P34, G36, D43 and P44. An acetylation step was systematically introduced after each amino-acid coupling in order to eliminate truncation products. For manual synthesis, Fmoc protecting groups were cleaved with 4-methylpiperidine in DMF (1:5 v/v, 1 × 1 min + 2 × 7 min) except for the final coupling of PEG motifs on N-terminally linked dimeric peptides for which the deprotection was performed using 4-methylpiperidine in NMP (1:5 v/v, 1 × 1 min + 4 × 5 min). Couplings of Fmoc-protected amino acid were carried out in the presence of HBTU/HOBt and DIPEA (5:5:10 eq, 1h30min). C-terminally linked dimeric peptides were obtained by generating two points of chain elongation with a Fmoc-Lys(Fmoc)-OH amino acid, whereas N-terminally linked dimeric peptides were obtained by coupling a 7:3 mixture of Fmoc-Lys($N_3$)-OH and pentynoic acid followed by copper(I)-catalysed azide-alkyne cycloaddition in DMF/4-methylpiperidine (8:2) with CuI (5 eq), ascorbic acid (10 eq) and aminoguanidine (10 eq). N-free peptide resins were derivatised with acetyl groups or biotin. Acetylation was performed using 7 mL of a solution of 0.15 M acetic anhydride and 0.15 M pyridine in DMF (1 h 30 min). Biotinylation was performed by first coupling Fmoc-TTDS-OH (3 eq) then biotin (3 eq). Both couplings were mediated by HBTU/ HOBt and DIPEA (3:3:6 eq, 1 h 30 min). Acetylated peptides were used in living cells and for SPR experiments. Biotinylated peptides were used for affinity isolation from rat brain lysates and in SPR experiments.. Peptides were cleaved from the resin for 3 h using TFA/H₂O/TIPS (95:2.5:2.5) with mild orbital shaking and precipitated with cold diethyl ether. All crude peptides were resuspended in H₂O containing a few drops of DMF and purified in a Waters 1525 semi-preparative Reverse-phase high performance liquid chromatography (RP-HPLC) system equipped with a UV/Vis detector and a YMC C18, ODS-A/120, 250 × 20 mm. Purification was performed using a standard gradient of 5% MeCN containing 0.1% TFA for 5 min followed by an appropriate gradient of MeCN containing 0.1% TFA in H₂O for 40 min (1 mL min⁻¹). UV detection was performed at 220 and 280 nm. All peptides were obtained at more than 90% purity as judged by analytical RP-HPLC with the exception of dD15-N (84.8% purity). Masses were determined by MALDI-TOF spectroscopy in an UltraFlex III TOF-TOF (Bruker Daltonics). Peptide concentrations were determined by absorbance

measurements at 280 nm in a UV-visible spectrophotometer (UV-1800, Shimadzu) ($\varepsilon(Y) = 1490$ M⁻¹ cm⁻¹). Peptides were lyophilised and stored at −80 °C until use.

**Affinity isolation assays**. Brains were obtained from adult (2–3 months old) Sprague-Dawley rats raised in the animal facility of Bordeaux University B 33 063 917. Animals were killed by decapitation after isoflurane anaesthesia (5%, 3 min), in accordance with the European 2010/63/EU directive and approved by the Bordeaux University Ethics Committee (CE50). Frozen brains (2 × ~1.5 g) were thawed in 20 mL ice cold modified RIPA buffer (50 mM Tris pH 7.5, 150 mM NaCl, 0.1% SDS, 0.5% sodium deoxycholate, 1% NP-40, 1 mM EDTA) containing a protease inhibitor mixture (1:1000; Protease Inhibitor Cocktail set III; Calbiochem) for about 5 min and cut into small pieces. The tissues were homogenised using a glass/teflon homogeniser. Homogenates were centrifuged at 7500 × g for 25 min at 4 °C to remove cell debris. The supernatant was aliquoted and stored at −80 °C until the affinity-based isolation (pull-down) experiments were performed. Streptavidin-coated beads (Dynabeads M-280, Life Technologies) were washed three times and incubated for 15 min at room temperature (RT) in modified RIPA buffer supplemented with 0.1% BSA. Rat brain lysates were incubated with the biotinylated peptide (or biotin as a negative control) for 10 min at RT before addition of the beads and further incubation for 5 min at RT. Beads were washed five times in RIPA buffer and transferred into new eppendorf tubes. For elution, acetylated ligands were added in excess to the bead suspension and incubated for 5 min at RT. The supernatant was kept for proteomics analysis and electrophoresis followed by silver staining after addition of fresh 6× sample buffer (ProteoSilver Silver Stain Kit, Sigma-Aldrich).

**Proteomics analysis**. Samples were solubilized in Laemlli buffer and were deposited in triplicate onto SDS-PAGE. Separation was stopped once proteins have entered resolving gel. After colloidal blue staining, bands were cut out from the SDS-PAGE gel and subsequently cut in 1 mm × 1 mm gel pieces. Gel pieces were destained in 25 mM ammonium bicarbonate 50% MeCN, rinsed twice in ultrapure water and shrunk in MeCN for 10 min. After MeCN removal, gel pieces were dried at room temperature, covered with the trypsin solution (10 ng/μl in 50 mM NH₄HCO₃), rehydrated at 4 °C for 10 min, and finally incubated overnight at 37 °C. Spots were then incubated for 15 min in 50 mM NH₄HCO₃ at room temperature with rotary shaking. The supernatant was collected, and an H₂O/MeCN/ HCOOH (47.5:47.5:5) extraction solution was added onto gel slices for 15 min. The extraction step was repeated twice. Supernatants were pooled and dried in a vacuum centrifuge to a final volume of 25 μL. Digests were finally resuspended in 25 μl of formic acid (5%, v/v) and stored at −20 °C.

Peptide mixture was analysed on a Ultimate 3000 nanoLC system (Dionex, Amsterdam, The Netherlands) coupled to a nanospray LTQ-Orbitrap XL mass spectrometer (ThermoFinnigan, San Jose, CA). Ten microliters of peptide digests were loaded onto a 300-μm-inner diameter × 5-mm C₁₈ PepMap™ trap column (LC Packings) at a flow rate of 30 μL/min. The peptides were eluted from the trap column onto an analytical 75-mm id × 15-cm C18 Pep-Map column (LC Packings) with a 4–40% linear gradient of solvent B in 35 min (solvent A was 0.1% formic acid in 5% MeCN, and solvent B was 0.1% formic acid in 80% MeCN). The separation flow rate was set at 300 nL/min. The mass spectrometer operated in positive ion mode at a 2-kV needle voltage. Data were acquired in a data-dependent mode. Mass spectrometry (MS) scans (m/z 300–1700) were recorded at a resolution of $R = 70,000$ (@ m/z 400) and an AGC target of $5 \times 10^5$ ions collected within 500 ms. Dynamic exclusion was set to 30 s and top 6 ions were selected from fragmentation in CID mode. MS/MS scans with a target value of $1 \times 10^4$ ions were collected in the ion trap with a maximum fill time of 200 ms. Additionally, only +2 and +3 charged ions were selected for fragmentation. Others settings were as follows: no sheath nor auxiliary gas flow, heated capillary temperature, 200 °C; normalized CID collision energy of 35% and an isolation width of 3 m/z.

Data were searched by SEQUEST through Proteome Discoverer 1.4 (Thermo Fisher Scientific Inc.) against a subset of the 2018.01 version of UniProt database restricted to *Rattus norvegicus* Reference Proteome Set (29,961 entries). Spectra from peptides higher than 5000 Da or lower than 350 Da were rejected. The search parameters were as follows: mass accuracy of the monoisotopic peptide precursor and peptide fragments was set to 10 ppm and 0.6 Da respectively. Only b- and y-ions were considered for mass calculation. Oxidation of methionines (+16 Da) and carbamidomethylation of cysteines (+57 Da) were considered respectively as variable and fixed modifications. Two missed trypsin cleavages were allowed. Peptide validation was performed using Percolator algorithm[55] and only high confidence peptides were retained corresponding to a 1% False Positive Rate at peptide level.

Raw LC-MS/MS data were imported in Progenesis QI for Proteomics 2.0 (Nonlinear Dynamics Ltd, Newcastle, U.K). Data processing includes the following steps: (i) Features detection, (ii) Features alignment across the six samples, (iii) Volume integration for 2–6 charge-state ions, (iv) Normalization on feature median ratio, (v) Import of sequence information, (vi) Calculation of protein abundance (sum of the volume of corresponding peptides), (vii) A statistical test was performed and proteins were filtered based on p-value < 0.05. Noticeably, only non-conflicting features and unique peptides were considered for calculation at protein level. Quantitative data were considered for proteins quantified by a minimum of two peptides.

**SH3 domain production**. SUMO-amphSH3 and GST-amphSH3 were generated by subcloning murine Ampiphysin1 SH3 domain residues 607–686) from pAmph1-mCherry (kind gift from C. Merrifield, Addgene 27692) using BamHI and XhoI restriction sites into a modified pET-SUMO bacterial expression vector (Life Technologies) incorporating a TEV cleavage site and a multiple cloning site following the SUMO tag or a modified pGEX-4T-2 vector incorporating a TEV cleavage site before the gene of interest, using the following primers: 5′-ggatc-cactcaggaactgcctcctggc-3′ (forward) and 5′-ctcgagttactccaggccgccgcgtgaagtt-3′ (reverse). The biotinylated SUMO-amphSH3 was obtained by subcloning the gene from SUMO-amphSH3 using BsrGI and XhoI restriction sites into a homemade vector derived from pET-24a that presents a biotin acceptor peptide (Avitag) and a His10 tag prior to the gene of interest, using the forward primer 5′-ggcctgtacaa-gatgtcggactcagaagtc-3′ and the same reverse primer as above. The recombinant SH3 domains (SUMO and GST fusions) were produced in BL21-CodonPlus (DE3)-RIPL E. coli cells by auto-induction (Agilent, 230280) at 16 °C and purified by affinity (Ni-NTA resin) and size exclusion chromatography as previously described[39]. The biotinylated SUMO-amphSH3 fusion was produced in BL21 (DE3) (source ThermoFisher Scientific, C600003) E. coli cells in presence of a plasmid encoding for BirA in a pACYC-Duet-1 vector and biotin (50 µM) by auto-induction at 16 °C and purified as above. Proteins were stored at −80 °C in SPR running buffer until use. Analytical size exclusion chromatography were performed on a HiLoad Superdex200 Increase 10/300 GL column with PBS + 0.01% Tween-20 as an eluent at a flow rate of 0.75 mL min⁻¹.

**ELISA assay**. ELISAs were performed against SUMO-amphSH3. Nunc-Immuno MaxiSorp 96-well plates (ThermoScientific) were coated overnight at 4 °C with 100 µL of a 30 nM SUMO-amphSH3 solution in NaHCO₃ (50 mM, pH 9.6). Wells were then blocked with 200 µL of blocking buffer (PBS, 0.5% (wt/vol) biotin-free BSA) for 1 h at room temperature. After removing the blocking buffer, wells were washed four times with PT buffer (PBS with 0.05% tween-20). Biotinylated peptide dilutions were performed in PT buffer and 100 µL of each dilution was added to the ELISA plates and incubated on a rocking shaker for 1 h at room temperature. Following 16 washes with PT, 100 µL of horseradish-peroxidase-conjugated streptavidin (Jackson ImmunoReaserch) diluted at 1:3000 in PT) was added to each well. Samples were incubated on a rocking shaker for 30 min, and were then washed 6 times with PT and 4 times with PBS. Next, 100 µL of 1-Step Ultra (3,3′,5,5′-tetramethylbenzidine) TMB-ELISA was added and the reaction was stopped with 100 µL 1 M H₃PO₄. Absorbance was read at 450 nm on a Multiskan GO plate reader and corrected for absorbance at 800 nm as well as for no-specific binding of the peptides by subtracting the response in absence of SH3 domain. Data from independent run were normalized to the maximum values of the D54-dyn2 peptide and fitted using the One site binding hyperbola formula from GraphPad Prism v7.02.

**SPR measures and analysis**. Experiments were performed at 25 °C with a Bia-core™ X100 or T200 apparatus (Biacore™, GE healthcare Life Sciences, Uppsala, Sweden). The experiments were performed on biotin CAPture kit sensor chips (Biacore™). 50–90 RU of biotinylated peptides were immobilised by injecting solutions prepared at 5–10 nM in running buffer (10 mM sodium phosphate buffer, pH 7.4, 150 mM sodium chloride and 0.01% Tween-20). Importantly, for all experiments, density of the monovalent D15 peptide (bound RU) was about twofold superior to the ones of divalent or D44 peptides to rule out artefact due to high surface density of the ligand. One flow-cell was left blank and used as a reference. For single cycle kinetics experiments, captured targets and analyte samples were prepared in running buffer and injected at 5 and 30 µl min⁻¹ respectively. The capture of streptavidin and regeneration of the functionalised surface was achieved following the manufacturer's recommended protocols. Sen-sorgrams were double-referenced using Biacore evaluation software (Biacore). No points were removed prior to data fitting. Spikes still present after the double-referencing process had no influence on the analysis. The kinetic data were ana-lysed using a 1:1 Langmuir binding model of the BIAevaluation software with the bulk refraction index (RI) kept constant and equal to 0 and the mass transfer constant (tc) kept constant and equal to 10⁸. In the case of fast association and dissociation rate constants that prevented reliable kinetics analysis, the curves were thus exploited using equilibrium analysis to obtain the dissociation equilibrium constants in GraphPad Prism (GraphPad Software v7.02 for Windows) by fitting the data with the one-site binding hyperbola formula

$$Y = \frac{B_{max} * X}{K_D + X} \qquad (1)$$

where $Y$ is the response (in RU), $B_{max}$ is the response value at maximal binding, $K_D$ is the affinity constant to be determined and $X$ is the protein concentration.

**Dynamin2 mutant design**. Dynamin 2 (human dynamin2bb according to the nomenclature of ref. [56]) plasmids were based on the dyn2-mCherry construct from Taylor et al. 2011 (Addgene 27689). We first generated dyn2-GFP by sub-cloning dyn2 into pEGFP-N1 vector with BglII/EcoRI sites. To generate GFP-dyn2, we amplified its ORF by PCR with the following primers: tcgtaacaactccgcccc (forward) ggcctcgagtcagtcgagcagggatggcgtc (reverse) and sub-cloned it into pEGFP-C1 with

BglII/XhoI sites. We then generated dyn2-GFP and dyn2-mCherry constructs in which the GFP/mCherry tag is relocated before the PRD (residue 740) with linkers composed of six residues (GTSGSS). We first generated an unlabeled dyn2 plasmid by subcloning it from the dyn2-GFP and GFP-dyn2 plasmids with Nhe/PmlI sites. We then amplified EGFP/mCherry from their respective vectors with primers catcagcaccagcactgtgggaacctccggaggctctatggtgagcaagggcgagg (forward) and cgggggtacaggcgttggaagagcctccggaggttccttgtacagctcgtccatggc (reverse) and the unlabeled dyn2 plasmid with primers tccacgcctgtacccccg (forward) and cacagtgctggtgctgatg (reverse), and generated the final plasmids with In-Fusion HD cloning kit (Clontech). Dyn2-GFP-ΔCter was generated by re-introducing EGFP into the dyn2-GFP plasmid with NheI/BsrGI sites, introducing a stop codon after the EGFP. The dyn2-GFP PRD mutants (ΔCter, A_mut, B_mut, C_mut AC_mut and ABC_mut) were generated by inserting the artificially synthesized PRD (Eurofins) containing these mutations into the dyn2-GFP construct by In-Fusion. All con-structs were verified by sequencing.

**Cell culture and transfection**. NIH 3T3 cells (ECACC 93061524) were cultured in DMEM supplemented with 1% sodium pyruvate, 1% glutamax and 10% fetal calf serum (Invitrogen). Cells were incubated at 37 °C in 5% CO₂ until use and sub-cultured every 2 to 4 days for maintenance up to passage 20. Cells were transfected with 1.5 µg of TfR-SEP plasmid 24 h prior to imaging and patch clamp. Care was taken to find the right balance between sufficient fluorescent signal to allow for proper imaging and reasonable overexpression to diminish potential saturation artefacts. Best results were obtained using Fugene 6 (Promega) as the transfection reagent but Lipofectamine 2000 (Invitrogen) has also been used in this study due to a temporary arrest of Fugene 6 production by the provider. All transfections were performed according to the manufacturers' recommendations. Four to eight hours after transfection, cells were seeded at a density of ~70,000 cells ml⁻¹ on 18 mm glass coverslips coated with poly-L-lysine (3 min application of 0.1 mM polylysine (Sigma P2636) in borate buffer, pH 8.0 followed by a quick rinse in PBS) and used the next day.

The MEF cell line obtained from tamoxifen inducible triple dynamin conditional mice (dynamin TKO cells[25]) was a kind gift from Pietro De Camilli (Yale School of Medicine, New Haven, CT). Cells were cultured in DMEM/ F12 supplemented with 1% sodium pyruvate, 1% glutamax and 10% fetal calf serum (Invitrogen). Dynamin TKO was induced by the application of 3 µM Hydroxytamoxifen (OHT, Sigma) for 2 days, then 0.3 µM for 4 more days before transfection. Cells were transfected by nucleofection with the kit MEF 1 and the program MEF T20 (Lonza, Switzerland) by mixing 3 µg of DNA (dynamin construct and TfR-SEP if applicable) with 10⁶ cells. After electroporation, cells were plated on 4–6 18 mm glass coverslips coated with fibronectin (50 nM, Sigma) and equilibrated in culture medium. Cells were used one day after for transferrin uptake assay or imaging. Complete dynamin knock-out and dyn2-GFP re-expression was verified by Western Blot (WB) with a pan dynamin antibody (Santa Cruz sc-6401, dilution 1/200). Transfection efficiency was estimated by counting the cells with DAPI staining and the transfected cells with dyn2-GFP fluorescence. It was 46 ± 9% in three separate experiments. Therefore, because the amount of expression of dyn2-GFP-WT in TKO cells estimated by WB was similar to the one in untreated (WT) cells (Fig. 3b), we conclude that TKO cells transfected with dyn2-GFP-WT have about twofold overexpression of dynamin compared to WT cells. On the other hand, there was in the range of expression obtained here no correlation between the level of dyn2-GFP-WT and A568-Tfn uptake (Supplementary Fig. 4B). In later control experiments (Supplementary Figs. 1C and 2), we have used another electroporation device (Neon, ThermoFisher) which yielded a slightly smaller fraction of transfected cells (22 ± 7%, n = 8) than the Lonza device.

SK-MEL-2 cell clone Ti95 (Dnm2-GFPᵉⁿ all edited) was a kind gift from David Drubin (University of California, Berkeley, CA). Cells were cultured in DMEM/ F12 supplemented with 1% sodium pyruvate, 1% glutamax and 10% fetal calf serum (Invitrogen). Cells were plated on poly-L-lysine coated 18 mm glass coverslips and imaged the next day.

BSC-1 cells (ECACC 85011422) were cultured in DMEM supplemented with 1% sodium pyruvate, 1% glutamax and 10% fetal calf serum (Invitrogen). Cells were incubated at 37 °C in 5% CO₂ until use and subcultured every 2 to 4 days for maintenance up to passage 20. Cells were transfected with 2 µg of FLAG-tagged β2 or β1 adrenergic receptor DNA (kind gifts of Dr Aylin Hanyaloglu, London UK) using Fugene 6 (Promega) and seeded at a density of ~70,000 cells ml⁻¹ on 18 mm glass coverslips 4–8 hours after transfection.

**Live cell imaging**. Imaging was performed at 34–37 °C on an Olympus microscope (Olympus IX71 or IX83) equipped with an inverted Apochromat ×60 oil objective of numerical aperture 1.49 in combination with a ×1.6 magnifying lens; or with a ×100 objective of numerical aperture 1.49 (Olympus). Samples were excited in TIRF illumination (Olympus IX2RFAEVA illuminator) with a 473 nm laser source (Cobolt). Emitted fluorescence was filtered with a 525/50 m filter (Chroma Tech-nology). Images were acquired by an electron multiplying charge coupled device camera (EMCCD QuantEM:512SC, Roper Scientific) controlled by the software MetaVue 7.8 (Molecular Devices). Two-colour imaging was performed with a co-aligned 561 nm laser (Cobolt) with the same TIRF angle as the 473 nm laser. Emitted fluorescence was filtered with a 605/52 m filter (Chroma Technology). Red and green fluorescence images were obtained 200 ms apart using a filter wheel

(Lambda 10-3, Sutter Instruments). To correct for x/y distortions between the two channels, images of fluorescently labelled beads (Tetraspeck, 0.2 μm; Invitrogen) were taken before each experiment and used to align the two channels. Time lapse images were acquired at 0.5 Hz with an integration time of 100 ms.

**ppH assay**. Fast solution exchange was controlled by electro-valves (Lee company) connected to a theta glass pipette (TG150, Harvard Apparatus) pulled to ~100-μm tip size using a vertical puller (Narishige, World Precision Instruments) placed above the recorded cell. Cells were alternatively perfused with HEPES or MES buffered saline solutions (HBS/MBS) containing (in mM): 135 NaCl, 5 KCl, 0.4 MgCl$_2$, 1.8 CaCl$_2$, 1 D-glucose and 20 HEPES or MES. The pH was adjusted to 7.4 or 5.5 using NaOH and osmolarity to 310–315 mOsm L$^{-1}$ with NaCl.

**Combined patch-clamp and imaging**. Cells were patched with pipettes made of borosilicate glass (GC150F, Harvard Apparatus) and pulled with a vertical Narishige puller (World Precision Instruments). The solution contained (in mM): 130–140 KCH$_3$SO$_3$, 1 EGTA, 0.1 CaCl$_2$, 10–20 HEPES, 4 ATP (sodium or magnesium salt), 0.4 Na-GTP (or GTPγS when indicated), 5 phosphocreatine and 3 sodium ascorbate. pH was adjusted to 7.2 using KOH and osmolarity to ~300 mOsm L$^{-1}$. Sodium ascorbate was omitted in earlier recordings but was added for better stability. Chloride ions either came from 2 mM KCl, 10 mM NaCl or 4 mM MgCl$_2$ (when Na-ATP was used) in various trials. All salts were purchased from Sigma-Aldrich. Cells were voltage-clamped at −60 mV with an EPC10 amplifier (HEKA) with series resistance and cell capacitance estimation every 4 s in the whole cell mode. The digital triggers of the amplifier were used to drive the electrovalves for pH changes. Cell viability and dialysis efficiency were made sure of by monitoring three parameters throughout the recording using the Patchmaster software (HEKA): series resistance should remain below 10 MΩ and holding current should remain above −100 pA. If either of these criteria was not respected (except holding current for recordings in GTPγS, see Supplementary Fig. 3D), the recording was not considered for analysis. On a given experimental day at least one control recording (i.e. without any peptide) was performed to ensure recording quality. All these recordings were pooled in the 'control' condition ($n = 63$ on 42 experimental days). The nature of the peptide tested was unknown from the experimenter until the end of the recordings and analysis.

**Image analysis and data representation**. The recruitment of dyn2-GFP or amphiphysin-mCherry were analysed by segmentation and tracking (multi-dimensional image analysis (MIA) developed for Metamorph by JB Sibarita, Bordeaux, France) as described before[57]. Briefly, each image was segmented with a B-spline wavelet decomposition of scale 4 pixels and with a fixed threshold of eight times local standard deviation. Objects that could be tracked for more than three frames (6 s) were retained for further analysis with scripts written in Matlab 2017. The fluorescence value at each time point represents the average intensity in a 2 pixel (300 nm) radius circle around the centre of mass of the object to which the local background intensity is subtracted. This local background is estimated in an annulus (5 pixels outer radius, 2 pixels inner radius) centred on the region to be quantified as the average intensity of pixel values between the 20th and 80th percentiles (to exclude neighbouring fluorescent objects). The frame of maximum intensity is used as the reference (time 0) for aligning and averaging all traces. Before (resp. after) tracking of an object the fluorescence is measured at the location of the first (resp. last) frame with the object tracked. Two colour alignment was performed using an image of beads (TetraSpeck microspheres, ThermoFisher T7280) taken before the experiment.

The detection, tracking, and quantification of endocytic events was performed using scripts developed previously and updated in Matlab 2018a (Mathworks)[4]. Images were segmented according to the pH (7.4 or 5.5) into two movies. These two movies were segmented using MIA with fixed optimized parameters. Objects appearing in pH 5.5 images were then considered *candidate* CCVs if they were visible for more than three frames (8–12 s), if their signal to noise ratio was high enough, and if a cluster pre-existed where they appeared for at least five frames (20–24 s) in pH 7.4 images (Supplementary Fig. 2B). Importantly, this automated method of detection had been characterised as missing ~30% of all events but with a low percentage of false positives (~20%)[4,28], making it ideal for comparative studies over large datasets. In rescue experiments in dynamin TKO cells all events were reviewed by eye. For the experiments of peptide dialysis, to handle the large dataset without reviewing all the events by eye, we developed a Support Vector Machine (SVM) type neural network by supervised learning. We used a dataset of 26 recordings obtained in 6 experimental sessions with 9631 candidate events reviewed by two different operators (MR and DP), leading to 7763 bona fide endocytic events. This gives $19.1 \pm 0.1\%$ of false positives, a percentage virtually identical to previous estimates[4,28]. The inputs of the SVM were sets of $15 \times 15$ pixel images centred on the candidate event, 5 frames before and 6 frames after the candidate event is detected, i.e., $15 \times 15 \times 11 = 2475$ input values (pixel values) per candidate event. The output was binary, 'accepted' or 'rejected' (Supplementary Fig. 2C, D). Once trained, the SVM model could sort correctly all events used for training. We tested it further with 7 recordings with 1224 candidate events sorted by the user in 1124 bona fide events. The SVM sorted 1095 bona fide events ($98.4 \pm 2.0\%$ of the number of events accepted by the user). Finally, $93.5 \pm 1.1\%$

were the same as the events accepted by the users. We conclude from this analysis that the SVM trained by the 26 recordings is able to discriminate bona fide events with the same accuracy as a human operator, and probably with better consistency. In conclusion, we developed a fully automated analysis workflow to assess quantitatively the endocytic activity of cells in real time.

The cumulative frequency plots presented in Fig. 5b, c and Supplementary Fig. 6 were obtained by plotting event detections against time. For each recording, the frequency of events in both cell attached (resp. control) and whole cell (resp. treatment) recordings were normalised to the basal frequency of the cell as monitored in cell attached configuration. In other words, the curves were normalised so that the number of detections at the 70th frame of cell attached recording (280 s) would be set to 100. In this way, several cells submitted to the same treatment could be pooled into a single curve despite a great variability in their basal endocytic activity. The effect of a blocker was then measured as

$$\left( \frac{F_{WC}([8-10])}{F_{CA}([2-4])} \right) * 100 \qquad (2)$$

where $F$ is the frequency of events in the indicated time interval (in minutes) and the subscript is the recording mode in which it is being measured (CA: cell attached, WC: whole cell).

The quantification of TfR-SEP (at pH 7.4 and 5.5) and dyn2-mCherry was performed as in in the same manner using the coordinates of TfR-SEP at pH 5.5 (vesicles) and explained in detail in ref. [4]. Correction for bleed through was performed by minimizing the difference between the dyn2-mCherry fluorescence values at the two pH with a green to red bleed through factor. Finally, 95% confidence intervals for significant recruitment were obtained by shifting the real event coordinates within the cell mask 200 times and computed fluorescence on these shifted coordinates.

**Transferrin uptake assay and quantification**. TKO cells were starved in pre-warmed HBS for 10 min, then incubated in cold (4 °C) HBS containing 10 μg/ml of transferrin Alexa 568 (Tfn-A568, Thermo Fischer). Cells were then incubated at 37 °C for 5 min for endocytosis to occur. Afterwards, cells were washed in cold HBS, then in cold acetate buffer (pH 4.5) for 2 min to strip Tfn-A568 remaining on the cell surface, rinsed, and fixed with paraformaldehyde (4% in PBS, 37 °C) for 15 min. Cells were then rinsed several times in PBS and kept at 4 °C until imaging. Fixed cells were then imaged at the Bordeaux Imaging Center with a spinning disk confocal microscope (Leica DMI 6000 microscope (Leica Microsystems, Wetzlar, Germany) equipped with a confocal head Yokogawa CSU-X1 (Yokogawa Electric Corporation, Tokyo, Japan), and a QuantEM camera (Photometrics, Tucson, USA), controlled with Metamorph (Molecular Devices)). The laser and camera parameters kept constant for the whole set of conditions. A stack of six images separated by 0.5 μm encompassing the bottom part adhering to the coverslip was taken for each cell. We used the maximal projection of the z-stack for quantification. For each cell, we traced around the dyn-GFP image (when applicable) a mask of the cell, as well as a background region with no cell. In non-transfected cells the mask was traced around the Tfn-A568 image. We detected Tfn-A568 clusters by wavelet segmentation using custom written Matlab programs. For each cell we computed the average fluorescence, the number of clusters and the cell surface. These numbers were normalized to 100 for the control condition (no OHT induction) for each of four independent internalization assays. The distribution of dyn-GFP was scored for each cell as entirely punctuate (1), punctuate on a homogenous background (2), mostly homogenous with some puncta (3), or entirely homogenous (4) (see Fig. 1f for illustrations). All data processing and scoring was performed blind.

**β adrenergic receptor internalization assay**. Two days after transfection, cells were pre-treated (20 min, 37 °C) with either DMSO, Dyngo4a (30 μM, Abcam), myrD15 (10 μM, Tocris) or myrCtrl (10 μM, Tocris), incubated live with FLAG M1 antibody (Sigma F3040, dilution 1/200) (10 min, 37 °C), stimulated with Iso-proterenol (10 μM, 10 min, 37 °C), rinsed 3 times with PBS + Ca$^{2+}$ and fixed with paraformaldehyde (4% in PBS, 15 min, 37 °C). Fixed cells were incubated in blocking buffer (2% BSA in PBS + Ca$^{2+}$) for 1 h at RT, permeabilized (0.02% Triton-X100 in blocking buffer, 15 min, RT) and incubated with AlexaFluor555 anti-mouse secondary antibody (Thermo Fisher A31570, 1/1000) in the dark for 1 h at RT. Image acquisition and analysis was conducted following the same procedure as for Transferrin uptake assay and quantification described above.

**Statistical analysis**. Data is presented as average ± SEM unless otherwise stated. Differences between conditions were tested by a one-way ANOVA followed by Tukey's multiple comparison tests. Dyn2-GFP localization scores were compared with Kruskall–Wallis non parametric test followed by Dunn's multiple comparison tests. Variations of $C_m$ during whole cell recordings were tested with Student's paired $T$ tests. All values were calculated with Prism 7 (Graphpad). $P$ values are displayed in the Supplementary Table 3.

**Reporting summary**. Further information on research design is available in the Nature Research Reporting Summary linked to this article.

## Code availability

The Matlab programs that were used to analyze the ppH data are written for Matlab2018a with the following Matlab toolboxes: Image Processing, Wavelet, Statistics and Machine Learning. These programs are available at Matlab Central File Exchange as 72744-scission_analysis.

## Data availability

Data supporting the findings of this manuscript are available from the corresponding authors upon reasonable request. A reporting summary for this Article is available as a Supplementary Information file.

The mass spectrometry proteomics data have been deposited to the ProteomeXchange Consortium via the PRIDE[58] partner repository with the dataset identifier PXD015292.

The source data underlying Figs. 1e, f, 2b–d, g, 4b, e, 5d and Supplementary Figs. 1, 2 and 6 are provided as a Source Data file.

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

## Acknowledgements

We thank Christelle Breillat, Célia Michel and Natacha Retailleau (IINS) for help with plasmid constructs, Christopher Borcuk for help with Matlab programming, Pascal Desbarats (LaBRI, Talence) for counselling on the SVM for sorting scission events, Sébastien Marais (Bordeaux Imaging Center, part of the France BioImaging national infrastructure ANR-10-INBS-04) for help with the spinning disk confocal microscope. We also thank the Biochemistry and Biophysics Core Facility of the Bordeaux Neurocampus funded by the Labex BRAIN (ANR-10-LABX-43) and J. M. Blanc and Y. Ruffin for technical assistance as well as the Structural Biophysico-Chemistry platform (UMS3033/US001) of the Institut Européen de Chimie et Biologie (Pessac, France) for access to the T200 Biacore instrument and to Laetitia Minder for technical assistance. We thank Pietro De Camilli (Yale University) and David Drubin (UC Berkeley) for the gift of cell lines, and Aylin Hanyaloglu (Imperial College London) for plasmids. We thank Volker Haucke (FMP, Berlin) for critically reading the manuscript. This work was supported by the Centre National de la Recherche Scientifique (Interface program), the Fondation Recherche Médicale (FRM ING20101221208), the Agence Nationale pour la Recherche (CaPeBlE ANR-12-BSV5-005 and LocalEndoProbes ANR-17-CE16-0012) to D.P., the FRM, a pre-doctoral fellowship from the University of Bordeaux and a Labex BRAIN fellowship to M.R. and the European Research Council (advanced grants ADOS and Dyn-Syn-Mem) to D.C.

## Author contributions

M.R., M.S. and D.P. conceived the study and formulated the models. M.R. performed most of the ppH patch clamp experiments and analysed the corresponding data. T.N.N.V. performed most TKO cell experiments and analysed the corresponding data. D.G-B. synthesized and characterized the peptides and performed the peptide pull-down experiments. S.S. performed the experiments with the myrD15. L.C. performed some control experiments on TKO cells. I.G. produced and purified the SH3 domains. S.C. performed mass spectrometry data analysis. M.S. performed the SPR experiments and supervised all aspects of biochemical experiments. D.P. wrote the Matlab analysis programs, performed some ppH patch clamp and TKO live cell imaging experiments and analysed the live cell imaging experiments. M.R. and D.P. wrote the manuscript and all the other authors edited the manuscript.

## Competing interests

The authors declare no competing interests.
