## [Peer Review File · Nature Communications]

Reviewers' Comments:

Reviewer #1:

Remarks to the Author:

Rosendale et al in this Ms have studied the mechanism of dynamin recruitment to plasma membrane CCPs by endocytic SH3 domains. Using elegant rescue experiments with dynamin 2 variants carrying mutations in its proline-rich C-terminal domain (PRD) suggest that efficient dynamin 2 recruitment in cells requires multivalent interactions between two PRD motifs with dimeric SH3 domains of endocytic proteins. Further in vitro experiments using extended monomeric and designed dimeric PRD peptides confirm multivalency as an important principle for SH3 domain association. Finally, when dialyzed into cells dimeric peptides more potently inhibit CME in comparison to previously used monomeric variants.

This is a thoroughly executed study that provides interesting molecular insight into the mechanism of dynamin recruitment by endocytic SH3 domain proteins. Of importance, this work identifies dimeric dynamin 2-derived PRD peptides as potent inhibitors of dynamin-base endocytosis. That said I have a number of points that should be addressed prior to publication, both with respect to the mechanism of CME and the use of dimeric peptides as endocytosis inhibitors.

1. Although I find the CME inhibition data using the novel peptide variants compelling I feel that the mechanism by which they operate is not entirely clear. The tools generated in my view provide an ideal set of reagents to work out more precisely which endocytic SH3 domain proteins recruit dynamin to forming CCPs. For example, one could use the novel peptide reagents to determine their effect on dynamin complex formation in cells with various endocytic proteins of interest such as amphiphysin 1/2, endophilin A1-3, intersectin 1/2, CIN85, SNX9/18 etc in an at least semi-quantitative manner. Such experiments would yield important information on the role of these factors in recruiting dynamin to endocytic sites.

2. In light of the proposed reciprocal interactions between dynamin and SH3 domain-containing BAR proteins (Meinecke et al McMahon, JBC 2013) it would be interesting to analyze if application of these peptides to cells in addition to dynamin also impairs the recruitment of SH3 proteins such as endophilin A or amphiphysin.

3. Although dynamin plays an important role in CME it also drives membrane fission in other endocytic pathways. Do the novel dimeric peptides also inhibit clathrin-independent, dynamin-dependent endocytic uptake pathways such as FEME/ Shiga toxin internalization or caveolar uptake (one of them suffices) or can one consider them to be specific CME inhibitors?

4. An interesting aspect of this study is the observation that the peak fluorescence of dynamin 2 recruitment is reduced if the B peptide is mutated. This seems peculiar as previous work has claimed that only 26 molecules of dynamin are recruited to late-stage CCPs at peak, sufficient to form 1 to max 2 rings of dynamin. As these are average numbers I wonder whether only those events in which more than the peak average number of dynamin 2 molecules is recruited yield a productive fission reaction. Or do the authors think that even less than 26 dynamin 2 molecules suffice for fission? If so, this would greatly impact our understanding of dynamin-mediated membrane scission.

Minor points:

p. 6: "Under normal conditions" should perhaps be rephrased to physiological conditions or alike.

The delta PRD mutant referred to in the results is missing from the scheme in Fig. 1D.

p. 9: The authors state that with respect to Fig. 4G that 70-75kDa is the expected molecular weight for amphiphysin. Although correct in theory, amphiphysin due to the presence of a highly negatively charged stretch in its clathrin/ AP2 binding linker migrates at 115-125kDa in most SDS

PAGE systems. Whether the 75kDa band then truly is amphiphysin, remains questionable in my view

p.10: The authors refer to the D44 peptide in the text while Fig.5 says D45.

I suggest the interesting and compelling capacitance measurement shown in S4F to be moved to the main figure.

Reviewer #2:

Remarks to the Author:

The group of Dr. Perrais addresses an outstanding question in the field of endocytosis: How does dynamin interact with its SH3-binding partners during endocytosis? Specifically, this manuscript provides clear results showing the importance of multivalent contacts to increase the affinity of dynamin to SH3 domain containing proteins. Using the dynamin triple knock-out cells, they transfected cells with WT and several dynamin SH3-binding site mutants and found that disrupting only one SH3 binding site had minimal effect on endocytosis. In contrast, mutating all three binding sites, and to some extent two sites, caused a large reduction in transferrin uptake. This suggested all three sites are necessary or optimal endocytosis. The number of dynamin puncta in the cells also correlated with transferrin uptake. To achieve more accurate membrane localization of the proteins, TIRF was implemented and used to measure intensity of dynamin at sites of transferrin uptake. Peptides derived from the PRD of dynamin with one or more SH3 binding domains further illustrated that multi-binding domains enhance the interactions with SH3 domains, particularly if the SH3 domain is oligomerized. In addition, only peptides with 2 or more binding domains were successful in pulling down endocytic proteins in a pull-down assay. Further patch-clamp experiments nicely showed the direct effect of the peptides to endocytic events and as predicted the multivalent peptides had the greatest inhibition. Overall this is a well written manuscript that will be of great interest to the field of endocytosis.

Comments:

Based on these results there is a clear correlation between puncta and endocytic events suggesting the major role of the SH3 binding proteins is recruitment of dynamin and not the mechanical fission reaction. In the discussion, there is speculation of the role of the multivalent SH3 binding on the dynamin oligomer however, after recruitment the role of the SH3 domain may be insignificant and the BAR domain is the crucial feature of these proteins.

Minor points:

In figure 2, are the punctate vs homogenous detected by threshold levels and is so what was the level that defined punctate? This is defined in methods on pg 32 for the peptide experiment but not for the mutant experiments.

Comment on why is dyn2-GFP- Δ Cter is so bright in fig 2A and why this was not analyzed in fig 2B-E (background signal too high?).

Comment on the fact that dD15-C had lower affinity as shown in figure 4F, compared to dD15-N, but the greater effect in blocking CME, as shown in figure 5.

For the Bmut, the detection of less dynamin puncta could be due to a signal threshold and the defect in endocytosis may be due to unsuccessful endocytic events.

Briefly discuss the function of SH3bp1 and why it might be one of the highest pull down proteins?

In Figure 5, D45 should be D44?

It is hard to distinguish the red and orange in fig 4 and 5.

Reviewer #3:

Remarks to the Author:

This manuscript by Rosendale and colleagues examines the requirements of the amphiphysin SH3-binding motifs in the PRD of dynamin and the consequences of disruption of these motifs for endocytosis as assessed using a range of approaches and assays.

First, the authors show, by expressing various mutant constructs of dynamin-2-GFP-PRD in dynamin triple knock-out (TKO) cells, that the previously well-characterized major amph SH3 binding site (here named B and covered by previously reported peptide D15, is not the only determinant of dynamin 2 recruitment in clathrin-mediated endocytosis.

Using the ppH assay, the authors demonstrate a significant difference in CCV events in triple knockout cells when expressing the modified dynamin 2 and motif mutants.

Next, the authors switch to characterization of various peptides designed based on dynamin 1. Using amphiphysin SH3 constructs that are monomeric, dimeric and tetrameric, they show a marked enhancement in measured KD of binding to dynamin 1 D44 when amphiphysin is multimerized. Synthetic ligands, based on the dynamin I PRD and catenating D15 peptides using PEG linkers, behaved like D44.

Finally, the authors combine the ppH assay with patch clamping to allow dialysis of the various peptides into the TKO cells. Dialysis of D15 at 1mM had modest effects on CME but the catenated D15s had more significant effects.

The conclusion of the work is that efficient dynamin recruitment (and hence efficient endocytosis) requires multimeric binding partners. The PRD may accomplish this by presenting several amph SH3 binding sites.

Overall this is an interesting study that combines advanced peptide engineering with some elegantly adapted well-established assays to generate insight into the mechanisms of dynamin recruitment and its importance for endocytosis. However, some major concerns remain at this time that dampen my enthusiasm for supporting publication in its current form.

1) In Fig. 1B the authors show, using a pan-dynamin antibody, that expression of dynamin 2-GFP levels are comparable to those in TKO cells untreated with hydroxytamoxifen. In Fig. 1E, the authors quantify the number of transferrin clusters in TKO cells on expression of various PRD mutant and deletion constructs (on the backbone of the Dyn2-GFP-PRD construct). The authors should here assess expression levels in each case to eliminate the possibility that differences in Tfn clustering are due to alterations in expression levels of the expressed constructs. This also affects Fig. 1F.

2) The first part of the manuscript focuses on the dissection of, using various truncations and mutants, of the PRD of dynamin 2. The peptide work switches instead to peptides designed on the basis of the PRD of dynamin 1. The PRDs are the most divergent regions of the dynamins and the authors themselves point out that dynamin 1 lacks motif C, which was convincingly shown to contribute to rescue of CME by dynamin 2. Consequently, the authors should introduce peptides designed on the basis of dynamin 2 into the study, and especially for the characterization of binding with the monomeric, dimeric and tetrameric amph SH3 constructs.

3) The authors compare the functions of shorter versus longer peptides (eg D15 vs D44) and

suggest that the longer peptide is more efficient because it encompasses two potential amph SH3-interacting motifs (the dynamin 1 equivalents of A and B). To discount non-specific or context/environmental effects of the longer peptide, or even different binding motifs, a mutant disrupting A should be used as a control.

4) In Figs4 C-E, d15-C is not included in the sensorgrams and analyses. In 4F, d15-C is included but, surprisingly, has more modest effects than d15-N. Likewise, in the assay in Fig. 5, there appears to be a marked difference on dialysis of d15-N and d-15C into the cells. These peptides have the same sequence and differ only in their catenation. Hence the local environment and contexts surrounding the motifs apparently have significant effects on the results of the assays. To my mind, this weakens the results of comparisons between the various peptides, as these could also be sensitive to other non-specific context-dependent effects.

5) As an extension, the context of the PRD in the first part of the manuscript is significantly changed by interposing a GFP between the BSE and the PRD so the comparison with the gene-edited Dyn2-GFP (where the GFP follows the PRD rather than the other way round in the construct used for analysis) is perhaps not ideal.

6) Fig. 4G. The mass spec data provides only data for d15-N. To make the result meaningful, the data for D-15 should, minimally, be included. The authors state that no binding was observed with D-15. Hence no enrichment of any of the binding partners should be detected by mass spec. This data should be provided to assess this.

Minor

1) Can the authors include a GFP channel for Fig1C 1 too?

2) In Fig. 2C, I am intrigued by the apparent differences in event frequency between the gene-edited SKMEL cells (expressing dyn2-PRD-GFP) and the TKO cells after hydroxytamoxifen and expressing dyn2-GFP-PRD. Could the authors also express dyn2-PRD-GFP, as in the SKMEL cells, to rule out changes in event frequency due to moving the position of the GFP?

Reviewers' comments:

Reviewer #1 (Remarks to the Author):

Rosendale et al in this Ms have studied the mechanism of dynamin recruitment to plasma membrane CCPs by endocytic SH3 domains. Using elegant rescue experiments with dynamin 2 variants carrying mutations in its proline-rich C-terminal domain (PRD) suggest that efficient dynamin 2 recruitment in cells requires multivalent interactions between two PRD motifs with dimeric SH3 domains of endocytic proteins. Further in vitro experiments using extended monomeric and designed dimeric PRD peptides confirm multivalency as an important principle for SH3 domain association. Finally, when dialyzed into cells dimeric peptides more potently inhibit CME in comparison to previously used monomeric variants.

This is a thoroughly executed study that provides interesting molecular insight into the mechanism of dynamin recruitment by endocytic SH3 domain proteins. Of importance, this work identifies dimeric dynamin 2-derived PRD peptides as potent inhibitors of dynamin-base endocytosis. That said I have a number of points that should be addressed prior to publication, both with respect to the mechanism of CME and the use of dimeric peptides as endocytosis inhibitors.

We thank the reviewer for her/his evaluation of our work and the constructive remarks.

1. Although I find the CME inhibition data using the novel peptide variants compelling I feel that the mechanism by which they operate is not entirely clear. The tools generated in my view provide an ideal set of reagents to work out more precisely which endocytic SH3 domain proteins recruit dynamin to forming CCPs. For example, one could use the novel peptide reagents to determine their effect on dynamin complex formation in cells with various endocytic proteins of interest such as amphiphysin 1/2, endophilin A1-3, intersectin 1/2, CIN85, SNX9/18 etc in an at least semi-quantitative manner. Such experiments would yield important information on the role of these factors in recruiting dynamin to endocytic sites.

We show with the pull-down experiments on rat brain lysates (Figure 4G) that divalent peptides are able to bind to many SH3 domain containing proteins involved in endocytosis (Supplementary Table 2). However, we cannot rule out that these interactions are indirect. We agree with the reviewer that dissecting the effect of multivalency on dynamin complex formation with its various partners would be of the highest interest but we believe that the D15 sequence used in this study justifies our focus on amphiphysin. A large body of literature indeed has already characterized the degree of specificity of SH3 binding motifs in the PRD of dynamin to the SH3 domains of proteins involved in endocytosis. The PxRPxR motif found in the D15 patch of all dynamins, as well as synaptojanin and ataxin2, is highly specific for amphSH3 (Grabs et al. 1997; Landgraf et al. 2004). This SH3 domain contains one extra loop with acidic residues (Owen et al. 1998) that is absent from the SH3 domains of the other proteins involved in endocytosis. To test for a direct interaction of our peptides with amphiphysin, we have thus produced isolated amphiphysin SH3 domains in various degrees of oligomerization (Figure 4). Moreover, we show that the increased affinity of the dimeric peptides is due to the degree of oligomerization of the amphiphysin SH3 domains, with $K_d > 30 \mu\text{M}$ for monomeric SH3 and $K_d < 1 \mu\text{M}$ for multimeric ones (Figure 4), which is meant to mimic the situation of amphiphysin at the time of CCV formation in cells.

Ultimately, we demonstrate the specificity of the functional effect of divalent peptides with our live cell assay of CME.

2. In light of the proposed reciprocal interactions between dynamin and SH3 domain-containing BAR proteins (Meinecke et al McMahon, JBC 2013) it would be interesting to analyze if application of these peptides to cells in addition to dynamin also impairs the recruitment of SH3 proteins such as endophilin A or amphiphysin.

We have expressed amphiphysin-mCherry together with the dyn2-GFP mutant in dynamin TKO cells (new Figure 2, panels F,G). We show that the absence of dynamin recruitment does not affect the recruitment of amphiphysin (amplitude and frequency), in line with the results of Meinecke et al. (2013) on the effect of dyn1,2 KD on the recruitment of amphiphysin to CCPs. Measuring amphiphysin-mCherry recruitment with peptide infusion through a patch-clamp pipette would have given a similar result but was technically more challenging so we chose to perform the experiment shown in Figure 2F,G.

3. Although dynamin plays an important role in CME it also drives membrane fission in other endocytic pathways. Do the novel dimeric peptides also inhibit clathrin-independent, dynamin-dependent endocytic uptake pathways such as FEME/ Shiga toxin internalization or caveolar uptake (one of them suffices) or can one consider them to be specific CME inhibitors?

We have tested the uptake of several cargo proteins that are known to be clathrin independent. We have monitored the uptake of cholera toxin B and shiga toxin B but they were only partially sensitive to inhibition by a known dynamin inhibitor, dyngo4a (McCluskey et al., 2013; Renard et al., 2015) so we tested another cargo that is strictly dependent on dynamin function. The internalization of the G-protein coupled receptor β 1 adrenergic receptor depends on dynamin but not clathrin in the so-called FEME pathway (Boucrot et al., 2015). We set up an internalization assay of β 1AR and the closely related β 2AR which is internalized through CME (Goodman et al., 1996) in BSC-1 cells (Supplementary Figure 6). In this new data, we show that Dyngo4a can fully block the internalization of both receptors, induced by the β -adrenergic specific agonist isoproterenol. In the present study, we had tested the effect of D15 derived peptides using cell dialysis with a patch clamp pipette combined with the ppH live cell imaging of endocytosis. We knew we would be able to monitor the internalization of SEP- β 2AR with the ppH assay (we did in Shen et al., 2014) but unfortunately, we failed to detect the formation of endocytic vesicles containing SEP- β 1AR. Perhaps this is because these vesicles form at the dorsal surface of lamellipodia (Boucrot et al., 2015) out of the evanescent field (the ppH assay uses TIRF microscopy). Therefore, we settled on an assay using fixed cells and a peptide that can penetrate cells without the use of a patch-clamp pipette. We have used myristoylated D15 that blocks endocytosis in neuronal cells (Fujii et al., 2017; Glebov et al., 2015). We first show that myrD15 partially blocks TfR endocytosis by CME in 3T3 cells as monitored with the ppH assay (Supplementary Figure 6A,B). Then we show that myrD15 also partially blocks β 2AR internalization but does not affect β 1AR internalization. We conclude from this new data that D15-derived peptides could be selective tools to interfere with CME, leaving other forms of dynamin dependent endocytosis such as FEME unaffected. In addition to Supplementary Figure 6 and its description in the main text, we have written a sentence in the discussion:

“Interestingly, we found that myrD15 partially blocks CME of TfR and β 2AR but has no effect on the internalization of β 1AR, which depends on dynamin and endophilin but not on clathrin and amphiphysin (Boucrot et al., 2015; Hak et al., 2018). D15-derived peptides could thus antagonize specific forms of

dynamain dependent endocytosis such as CME which critically depend on the interaction between dynamain and amphiphysin.”

4. An interesting aspect of this study is the observation that the peak fluorescence of dynamain 2 recruitment is reduced if the B peptide is mutated. This seems peculiar as previous work has claimed that only 26 molecules of dynamain are recruited to late-stage CCPs at peak, sufficient to form 1 to max 2 rings of dynamain. As these are average numbers I wonder whether only those events in which more than the peak average number of dynamain 2 molecules is recruited yield a productive fission reaction. Or do the authors think that even less than 26 dynamain 2 molecules suffice for fission? If so, this would greatly impact our understanding of dynamain-mediated membrane scission.

We have quantified dynamain fluorescence at the time of scission in cells transfected with dyn2-mCherry-WT and dyn2-mCherry-B_{mut} (new panel H in Figure 3). We did not find a clear minimal or threshold value which would be consistent with a minimal number of dynamain molecules at the time of scission. On the other hand, we found that the distribution of fluorescence values was narrower for cells transfected with dyn2-mCherry-B_{mut} than with the dyn2-mCherry-WT. This is consistent with the hypothesis that in the WT case, the number of dynamain molecules recruited at the time of scission could be in excess compared to the B_{mut} case.

Minor points:

p. 6: "Under normal conditions" should perhaps be rephrased to physiological conditions or alike.

We have rephrased this sentence: "Motif C is the prime target when available. However, when this site is not available, motif B appears to be stabilized enough to impart a partial rescue."

The delta PRD mutant referred to in the results is missing from the scheme in Fig. 1D.

In the ΔPRD mutant the whole PRD domain is missing, including the amino-terminal part, therefore we cannot represent this mutant alongside the other mutants since only the C terminal half of the PRD is represented in this panel.

p. 9: The authors state that with respect to Fig. 4G that 70-75kDa is the expected molecular weight for amphiphysin. Although correct in theory, amphiphysin due to the presence of a highly negatively charged stretch in its clathrin/ AP2 binding linker migrates at 115-125kDa in most SDS PAGE systems. Whether the 75kDa band then truly is amphiphysin, remains questionable in my view

We have performed a Western Blot on rat brain lysate with an anti-amphiphysin antibody (Santa Cruz sc30099) which shows that amphiphysin has an apparent molecular weight of 90 kDa. We have removed the sentence referring to the size of amphiphysin.

Figure: Western blot on rat brain lysate. Lane 1, ladder; lane 2 rat brain lysate.

p.10: The authors refer to the D44 peptide in the text while Fig.5 says D45.

It is indeed D44. We have corrected that.

I suggest the interesting and compelling capacitance measurement shown in S4F to be moved to the main figure.

We agree that the data in Supplementary Figure 4F is very interesting and in line with the endocytosis imaging data. However, we feel that this panel is easier to understand alongside the rest of the data on electrophysiology in this Supplementary Figure so we would prefer to leave it there.

Reviewer #2 (Remarks to the Author):

The group of Dr. Perrais addresses an outstanding question in the field of endocytosis: How does dynamin interact with its SH3-binding partners during endocytosis? Specifically, this manuscript provides clear results showing the importance of multivalent contacts to increase the affinity of dynamin to SH3 domain containing proteins. Using the dynamin triple knock-out cells, they transfected cells with WT and several dynamin SH3-binding site mutants and found that disrupting only one SH3 binding site had minimal effect on endocytosis. In contrast, mutating all three binding sites, and to some extent two sites, caused a large reduction in transferrin uptake. This suggested all three sites are necessary or optimal for endocytosis. The number of dynamin puncta in the cells also correlated with transferrin uptake. To achieve more accurate membrane localization of the proteins, TIRF was implemented and used to measure intensity of dynamin at sites of transferrin uptake. Peptides derived from the PRD of dynamin with one or more SH3 binding domains further illustrated that multi-binding domains enhance the interactions with SH3 domains, particularly if the SH3 domain is oligomerized. In addition, only peptides with 2 or more binding domains were successful in pulling down endocytic proteins in a pull-down assay. Further patch-clamp experiments nicely showed the direct effect of the peptides on endocytic events and as predicted the multivalent peptides had the greatest inhibition. Overall this is a well written manuscript that will be of great interest to the field of endocytosis.

We thank the reviewer for her/his very positive evaluation of our work and for her/his constructive comments.

Comments:

Based on these results there is a clear correlation between puncta and endocytic events suggesting the major role of the SH3 binding proteins is recruitment of dynamin and not the mechanical fission

reaction. In the discussion, there is speculation of the role of the multivalent SH3 binding on the dynamin oligomer however, after recruitment the role of the SH3 domain may be insignificant and the BAR domain is the crucial feature of these proteins.

We agree with the reviewer that our data provide evidence that multivalent interactions strongly affect dynamin recruitment (it appears that by regulating its probability of recruitment these interactions indirectly regulate its amount). However, we have tried in the discussion to highlight the importance of both the BAR and the SH3 domains of amphiphysin. The former allows amphiphysin and others to timely concentrate only once the neck is highly curved, while the latter, by reorganizing as multimers, provide a platform for stably recruiting dynamin via its multiple SH3-binding motifs. We thus think that BAR and SH3 domains respectively regulate dynamin recruitment at CCPs (as opposed to in the cytosol) in time and space.

Minor points:

In figure 2, are the punctate vs homogenous detected by threshold levels and is so what was the level that defined punctate? This is defined in methods on pg 32 for the peptide experiment but not for the mutant experiments.

We have detected clusters with the same wavelet segmentation and tracking used to detect endocytic vesicles. We have chosen a threshold (8 times a local background) that matches the visible clusters and kept this threshold constant for all experiments shown in Figure 2. We have clarified this point in the Methods section (page 35).

Comment on why is dyn2-GFP- Δ Cter is so bright in fig 2A and why this was not analyzed in fig 2B-E (background signal too high?).

The homogenous fluorescence is brighter in dyn2-GFP mutants than in the WT as shown in Figure 2B. The frequency of dyn-GFP- Δ Cter recruitment events was quantified in Figure 2C but it is way lower than for other conditions (0.001 ± 0.0007 vs 0.225 ± 0.029 for dyn-GFP- Δ Cter and dyn-GFP-WT, respectively). Because of this extremely low frequency, the amplitude and kinetics of these events was not calculated (Figure 2D,E).

Comment on the fact that dD15-C had lower affinity as shown in figure 4F, compared to dD15-N, but the greater effect in blocking CME, as shown in figure 5.

We now report in the Results that the K_D for Ntr-bSUMO-amphSH3 is slightly higher for dD15-C (34 nM) than for dD15-N (4.5 nM).

To comment on the apparent discrepancy between in vitro binding and effect on CME, we have added the following sentences in the Discussion: “dD15-N appeared to bind slightly more strongly to tetrameric amphSH3 than dD15-C (4.5 nM vs 34 nM). However, dD15-C was more potent in inhibiting CME in living cells than dD15-N, suggesting that the precise arrangement and dynamics of SH3 domains recruiting dynamin to nascent endocytic vesicles cannot be easily reproduced in vitro. In all cases, these tools will help to decipher the role of multimeric interactions for dynamic protein assemblies in cells, an emerging field in cell biology (Banani et al., 2017).”

For the Bmut, the detection of less dynamin punta could be due to a signal threshold and the defect in endocytosis may be due to unsuccessful endocytic events.

We agree with the reviewer that the decreased apparent frequency of recruitment events could be due to the fact that small amounts of dyn2-GFP-Bmut would remain below the detection threshold. This defect in dynamin recruitment would then lead to the defect in endocytosis. We indeed see that reduced amounts of dynamin reach CCPs on average, even at sites of successful endocytosis (Figure 3E,H). When the recruitment is too severely affected, endocytosis is inhibited.

Briefly discuss the function of SH3bp1 and why it might be one of the highest pull down proteins?

SH3bp1 in the rat genome has a human homolog, better known as SH3bp1/CIN85 (Cbl Interacting protein of 85 kD). It bears three SH3 domains, two of which (SH3A and C) interact with dynamin2 (Schroeder et al., 2010). We have added the following sentence (in red) to the Results part describing the proteins identified by mass spectrometry:

“...the co-precipitated proteins were highly enriched in CME related proteins and in particular in known interactors of dynamin such as **SH3bp1 (known as CIN85 in human)**(Schroeder et al., 2010) intersectin1-2, amphiphysin1 and amphiphysin2 (also known as BIN1), endophilinA1-2 and SNX18 (Supplementary Table 2). **Notably, all these interactors either contain multiple SH3 domains which bind dynamin (CIN85, intersectins) or a BAR domain which enables dimer formation (endophilins, amphiphysins, SNX18)(Peter et al., 2004).** Our *in vitro* data therefore support the idea that multimerisation of SH3 domains together with the presence of multiple SH3 binding motifs in dynPRD play an essential role in increasing the affinity of their interaction by avidity effects.”

In Figure 5, D45 should be D44?

Yes, it is D44. We have corrected that.

It is hard to distinguish the red and orange in fig 4 and 5.

We have made the orange color lighter for better distinction.

Reviewer #3 (Remarks to the Author):

This manuscript by Rosendale and colleagues examines the requirements of the amphiphysin SH3-binding motifs in the PRD of dynamin and the consequences of disruption of these motifs for endocytosis as assessed using a range of approaches and assays.

First, the authors show, by expressing various mutant constructs of dynamin-2-GFP-PRD in dynamin triple knock-out (TKO) cells, that the previously well-characterized major amph SH3 binding site (here named B and covered by previously reported peptide D15, is not the only determinant of dynamin 2 recruitment in clathrin-mediated endocytosis.

Using the ppH assay, the authors demonstrate a significant difference in CCV events in triple knockout cells when expressing the modified dynamin 2 and motif mutants.

Next, the authors switch to characterization of various peptides designed based on dynamin 1. Using amphiphysin SH3 constructs that are monomeric, dimeric and tetrameric, they show a marked enhancement in measured KD of binding to dynamin 1 D44 when amphiphysin is multimerized. Synthetic ligands, based on the dynamin I PRD and catenating D15 peptides using PEG linkers, behaved like D44.

Finally, the authors combine the ppH assay with patch clamping to allow dialysis of the various peptides into the TKO cells. Dialysis of D15 at 1mM had modest effects on CME but the catenated D15s had more significant effects.

The conclusion of the work is that efficient dynamin recruitment (and hence efficient endocytosis) requires multimeric binding partners. The PRD may accomplish this by presenting several amph SH3 binding sites.

Overall this is an interesting study that combines advanced peptide engineering with some elegantly adapted well-established assays to generate insight into the mechanisms of dynamin recruitment and its importance for endocytosis. However, some major concerns remain at this time that dampen my enthusiasm for supporting publication in its current form.

We thank the reviewer for her/his overall positive evaluation of our work and her/his constructive remarks. We have done our best to address the issues raised below.

1) In Fig. 1B the authors show, using a pan-dynamin antibody, that expression of dynamin 2-GFP levels are comparable to those in TKO cells untreated with hydroxytamoxifen. In Fig, 1E, the authors quantify the number of transferrin clusters in TKO cells on expression of various PRD mutant and deletion constructs (on the backbone of the Dyn2-GFP-PRD construct). The authors should here assess expression levels in each case to eliminate the possibility that differences in Tfn clustering are due to alterations in expression levels of the expressed constructs. This also affects Fig. 1F.

We have performed Western blot analysis of the eight dyn2-GFP constructs used in Figure 1 with an anti-GFP antibody. As shown in Supplementary Figure 1C, all constructs are expressed at similar levels. The occasional variations in the expression level likely reflects various degrees of transfection (estimated by comparing DAPI staining and GFP fluorescence $22 \pm 7 \%$, range 6-30 %, n = 8 on this dataset) with no general trend for any of the constructs.

2) The first part of the manuscript focuses on the dissection of, using various truncations and mutants, of the PRD of dynamin 2. The peptide work switches instead to peptides designed on the basis of the PRD of dynamin 1. The PRDs are the most divergent regions of the dynamins and the authors themselves point out that dynamin 1 lacks motif C, which was convincingly shown to contribute to rescue of CME by dynamin 2. Consequently, the authors should introduce peptides designed on the basis of dynamin 2 into the study, and especially for the characterization of binding with the monomeric, dimeric and tetrameric amph SH3 constructs.

We have added an experiment using a dynamin 2 derived peptide using an ELISA assay on immobilized amphSH3 at high density, which mimics oligomeric amphSH3. As predicted by our model, this dyn2 based peptide binds similarly to D44, and more strongly than D15 or a D44 peptide in which the arginine residue of the accessory motif was mutated (Figure 4B). We had initially used peptides based on the C-terminal part of dyn1 PRD to restrict our analysis to dimeric binding which simplifies the kinetic analysis of the SPR data.

3) The authors compare the functions of shorter versus longer peptides (eg D15 vs D44) and suggest that the longer peptide is more efficient because it encompasses two potential amph SH3-interacting motifs (the dynamin 1 equivalents of A and B). To discount non-specific or context/environmental effects of the longer peptide, or even different binding motifs, a mutant disrupting A should be used as a control.

As explained in point 2, we have tested the binding of a mutated D44 to immobilized amphSH3 with ELISA. Unlike D44, it does not bind strongly.

4) In Figs4 C-E, d15-C is not included in the sensorgrams and analyses. In 4F, d15-C is included but, surprisingly, has more modest effects than d15-N. Likewise, in the assay in Fig. 5, there appears to be a marked difference on dialysis of d15-N and d-15C into the cells. These peptides have the same sequence and differ only in their catenation. Hence the local environment and contexts surrounding the motifs apparently have significant effects on the results of the assays. To my mind, this weakens the results of comparisons between the various peptides, as these could also be sensitive to other non-specific context-dependent effects.

We now report in the Results that the K_D for Ntr-bSUMO-amphSH3 is slightly higher for dD15-C (34 nM) than for dD15-N (4.5 nM).

To comment on the apparent discrepancy between in vitro binding and effect on CME, we have added/modified the following sentences in the Discussion: “dD15-N appeared to bind slightly more strongly to tetrameric amphSH3 than dD15-C (4.5 nM vs 34 nM). However, dD15-C was more potent in inhibiting CME in living cells than dD15-N, suggesting that the precise arrangement and dynamics of SH3 domains recruiting dynamin to nascent endocytic vesicles cannot be easily reproduced in vitro. In all cases, these tools will help to decipher the role of multimeric interactions for dynamic protein assemblies in cells, an emerging field in cell biology (Banani et al., 2017).”

5) As an extension, the context of the PRD in the first part of the manuscript is significantly changed by interposing a GFP between the BSE and the PRD so the comparison with the gene-edited Dyn2-GFP (where the GFP follows the PRD rather than the other way round in the construct used for analysis) is perhaps not ideal.

To rule out any effect of the environment of dynPRD, we have compared the frequency of dyn2-GFP expressed in dynamin TKO cells for three positions of the GFP tag: N-terminal, before the PRD (the position of the constructs used in this study) and at the C-terminal position (Supplementary Figure 2). The frequencies, amplitudes and kinetics of recruitment events were not significantly different between the three constructs. Therefore, we conclude that the GFP tag does not perturb the PRD-SH3 interaction and the recruitment of dynamin.

6) Fig. 4G. The mass spec data provides only data for d15-N. To make the result meaningful, the data for

D-15 should, minimally, be included. The authors state that no binding was observed with D-15. Hence no enrichment of any of the binding partners should be detected by mass spec. This data should be provided to assess this.

We have reformulated the section to clarify our results. We now write: “Mass spectroscopy analysis of the pull-down material indicated that while D15 failed to show any significant enrichment in comparison to control conditions (absence of peptide), the material the co-precipitated proteins by dD15-N was highly enriched in CME related proteins and in particular in known interactors of dynamin...”

Essentially, in the same conditions with the monomeric D15 barely any protein could be detected by silver stained gel and proteomic analysis confirmed these results by showing no enrichment of any SH3 domain-containing protein (Max fold change < 1), with p values above 0.99 for all the detected relevant proteins, as shown in this Table.

Name	Peptide count	Unique peptides	Confidence score	Anova (p)	Max fold change
SH3 domain-containing proteins (known Dynamin partners)					
SH3kbp1/CIN85	30	29		0.99999926	0.205
Intersectin-2	20	20	221.06	0.99999926	0.944
Intersectin-1	ND	ND			
Amphiphysin1	ND	ND			
BIN1 (Amph2)	4	3	17.21	ND	0.163
Endophilin-A1	3	3	28.18	ND	0.001
Endophilin-A2	ND	ND			
SNX18	ND	ND			
SH3 domain-containing proteins (other)					
CD2-ap	22	21	345.81	ND	0.01
SH3d19	5	5	24.27	ND	0.232

Minor

1) Can the authors include a GFP channel for Fig1C 1 too?

The top left image of Figure 1C is of a cell that was not treated with hydroxytamoxifen and not transfected with GFP. Therefore we have no image in the GFP channel to provide.

2) In Fig. 2C, I am intrigued by the apparent differences in event frequency between the gene-edited SKMEL cells (expressing dyn2-PRD-GFP) and the TKO cells after hydroxytamoxifen and expressing dyn2-GFP-PRD. Could the authors also express dyn2-PRD-GFP, as in the SKMEL cells, to rule out changes in event frequency due to moving the position of the GFP?

We have compared the frequency of dynamin2 recruitment events with three positions of GFP. The frequency of dyn2-GFP recruitment events is not significantly different in the three conditions (Supplementary Figure 2), and still lower than the one measured in genome-edited SKMEL cells.

Reviewers' Comments:

Reviewer #1:

Remarks to the Author:

Rosendale et al in their revised Ms have nicely addressed all my questions and comments and I therefore enthusiastically support publication of this elegant work in Nat. Commun. I have two suggestions for minor improvements to the text before publication:

- 1) I suggest to include all calculated K_d values quoted in the main Ms text on p.10 into the respective legends in Fig 4. Currently, some values are reported (in panels B and E) while others (e.g. F) are not.
- 2) As there may be a small, yet statistically at this point insignificant effect of the myrD15 peptide on b1AR endocytosis I suggest to phrase the corresponding statements in the discussion a bit more cautiously to leave open the possibility that these or similar reagents may indeed perturb b1AR internalization at a different dose or in a different cell type.

REVIEWERS' COMMENTS:

Reviewer #1 (Remarks to the Author):

Rosendale et al in their revised Ms have nicely addressed all my questions and comments and I therefore enthusiastically support publication of this elegant work in Nat. Commun. I have two suggestions for minor improvements to the text before publication:

1) I suggest to include all calculated Kd values quoted in the main Ms text on p.10 into the respective legends in Fig 4. Currently, some values are reported (in panels B and E) while others (e.g. F) are not.

We have added the Kd values in the Figure legend.

2) As there may be a small, yet statistically at this point insignificant effect of the myrD15 peptide on b1AR endocytosis I suggest to phrase the corresponding statements in the discussion a bit more cautiously to leave open the possibility that these or similar reagents may indeed perturb b1AR internalization at a different dose or in a different cell type.

We have added the word “significantly” in the corresponding Results section (page 11), as well as in the Discussion (page 13).